# Selective BET bromodomain inhibition as an antifungal therapeutic strategy

Flore Mietton[1,2,*], Elena Ferri[3,*], Morgane Champleboux[1,*], Ninon Zala[1], Danièle Maubon[4,5], Yingsheng Zhou[3], Mike Harbut[6], Didier Spittler[2], Cécile Garnaud[4,5], Marie Courçon[1], Murielle Chauvel[7], Christophe d'Enfert[7], Boris A. Kashemirov[3], Mitchell Hull[6], Muriel Cornet[4,5], Charles E. McKenna[3], Jérôme Govin[1] & Carlo Petosa[2]

Invasive fungal infections cause significant morbidity and mortality among immunocompromised individuals, posing an urgent need for new antifungal therapeutic strategies. Here we investigate a chromatin-interacting module, the bromodomain (BD) from the BET family of proteins, as a potential antifungal target in *Candida albicans*, a major human fungal pathogen. We show that the BET protein Bdf1 is essential in *C. albicans* and that mutations inactivating its two BDs result in a loss of viability *in vitro* and decreased virulence in mice. We report small-molecule compounds that inhibit *C. albicans* Bdf1 with high selectivity over human BDs. Crystal structures of the Bdf1 BDs reveal binding modes for these inhibitors that are sterically incompatible with the human BET-binding pockets. Furthermore, we report a dibenzothiazepinone compound that phenocopies the effects of a Bdf1 BD-inactivating mutation on *C. albicans* viability. These findings establish BET inhibition as a promising antifungal therapeutic strategy and identify Bdf1 as an antifungal drug target that can be selectively inhibited without antagonizing human BET function.

[1] Institut de Biosciences et Biotechnologies de Grenoble, Laboratoire Biologie à Grande Échelle, Université de Grenoble Alpes, CEA, Inserm, 38000 Grenoble, France. [2] Institut de Biologie Structurale (IBS), Université de Grenoble Alpes, CEA, CNRS, 38044 Grenoble, France. [3] Department of Chemistry, Dana and David Dornsife College of Letters, Arts and Sciences, University of Southern California, University Park Campus, Los Angeles, California 90089, USA. [4] Laboratoire de Parasitologie-Mycologie, Institut de Biologie et Pathologie, CHU Grenoble Alpes, 38043 Grenoble, France. [5] Laboratoire TIMC-IMAG-TheREx, UMR 5525 CNRS, Université Grenoble Alpes, 38058 Grenoble, France. [6] California Institute for Biomedical Research, 11119 N Torrey Pines Rd, La Jolla, California 92037, USA. [7] Unité Biologie et Pathogénicité Fongiques, Institut Pasteur, INRA, 75015 Paris, France. * These authors contributed equally to this work. Correspondence and requests for materials should be addressed to C.E.M. (email: mckenna@usc.edu) or to J.G. (email: jerome.govin@inserm.fr) or to C.P. (email: carlo.petosa@ibs.fr).

nvasive fungal infections are a major global health concern, with ∼2 million cases and >800,000 deaths estimated annually worldwide[1]. *Candida* species such as *C. albicans* and *C. glabrata* are among the most significant human fungal pathogens, with invasive candidiasis yielding 30–40% mortality[2,3]. An increase in drug-resistant fungal strains and the limited repertoire of available drugs has led to an urgent need for novel therapeutic agents[1,4–6]. Promising results have emerged from the study of chromatin-interacting proteins as antifungal targets, including histone acetyltransferases and deacetylases[7,8]. Histone deacetylase inhibitors have weak antifungal activity when used alone but synergize with antifungal drugs such as azoles and echinocandins[8,9]. Deletion of either the histone deacetylase (*HST3*) or histone acetyltransferase (*RTT109*) regulating histone H3 Lys56 acetylation (H3K56ac) in *C. albicans* increases susceptibility to genotoxic and antifungal agents[10]. In a study of the Mediator complex subunit Med15, which interacts via its KIX domain with a transcription factor (Pdr1) implicated in pleiotropic drug resistance in *C. glabrata*, drug-resistant strains were re-sensitized to antifungals by a small-molecule inhibitor targeting the KIX domain–Pdr1 interface[11]. These findings point to an important role of chromatin-interacting proteins in fungal drug susceptibility.

Here we investigated an epigenetic reader module, the bromodomain (BD) from the bromo- and extra-terminal domain (BET) family, as a potential antifungal target in *C. albicans*. BET proteins are chromatin-associated factors that regulate transcription and chromatin remodelling[12]. Human BET family members are Brd2, Brd3, Brd4 and Brdt. BET proteins bind chromatin through their two BDs (BD1 and BD2), which specifically recognize histones acetylated on lysine residues. Whereas canonical BDs bind mono-acetylated histone peptides, BET BDs possess a wider ligand-binding pocket allowing them to recognize diacetylated peptides[13,14]. Small-molecule inhibitors, such as JQ1 and IBET, which selectively target BET BDs have been used to validate BET inhibition as a therapeutic strategy against cancer, cardiovascular disease, inflammatory disorders and other medical conditions, with several inhibitors currently in clinical trials[12,15–21].

The fungal BET protein Bdf1 has been characterized as a global transcriptional regulator in *Saccharomyces cerevisiae*, where it regulates over 500 genes[22]. *S. cerevisiae* Bdf1 (*Sc*Bdf1) associates with acetylated histones H3 and H4 (refs 22–24) and with the general transcription factor TFIID[25], and is a subunit of the SWR1 chromatin remodelling complex[24,26,27]. *Sc*Bdf1 is also important for chromatin compaction during sporulation[28] and for the salt stress response[29]. In addition to *BDF1*, *S. cerevisiae* possesses a second BET gene, *BDF2*, which is partly functionally redundant with *BDF1* (refs 24,25,30,31). Disruption of *BDF1* causes severe morphological and growth defects, while deletion of both *BDF1* and *BDF2* is lethal[22,23]. Point mutations that abolish ligand binding by *Sc*Bdf1 BD1 and BD2 cause growth and sporulation defects[24] and affect the majority of transcripts altered by disruption of the entire gene[22]. Many pathogenic fungal species (including *C. albicans*, *C. glabrata*, *Aspergillus fumigatus*, *Cryptococcus neoformans* and *Pneumocystis jirovecii*) lack *BDF2*, suggesting that inhibition of the sole BET family protein Bdf1 might significantly reduce viability and virulence[32].

Here we demonstrate that Bdf1 BD functionality is essential in *C. albicans* and identify small-molecule inhibitors that target Bdf1 BDs without inhibiting human BET proteins, establishing Bdf1 inhibition as a potential antifungal therapeutic strategy.

## Results

### *C. albicans* Bdf1 BDs bind multi-acetylated histone tails.
A phylogenetic tree of human and fungal BET proteins is shown in Fig. 1a. The BDs from *C. albicans* Bdf1 (*Ca*Bdf1) share 31–46% sequence identity with human BET BDs and 58–66% identity with those from *Sc*Bdf1 and hence are likely to bind multi-acetylated H3 and H4 tails (Fig. 1b). To verify this hypothesis, we purified recombinant *Ca*Bdf1 BD1 and BD2 and assessed binding to a microarray of human histone tail peptides bearing diverse post-translational modifications. While H2A and H2B tail sequences differ considerably between human and *C. albicans*, those of H3 and H4 are nearly perfectly conserved, justifying use of the array (Fig. 1c). As expected, *Ca*Bdf1 BDs showed weak binding to mono-acetylated peptides and stronger binding to peptides bearing two or more acetylation marks (Fig. 1d,e and Supplementary Data 1). Of the two BDs, BD1 exhibited more promiscuous binding, recognizing 14 distinct H3 and H4 acetylation patterns, versus only 7 for BD2. We replaced a conserved tyrosine residue in the binding pocket of each BD by phenylalanine, a mutation known to compromise ligand binding in *Sc*Bdf1 (refs 22,23) (Supplementary Fig. 1a,b). These two 'YF' mutations (Y248F and Y425F) abolished binding to all acetylated peptides, confirming interaction specificity (Fig. 1d,e). For both BD1 and BD2, the strongest binding was observed with an H4 peptide tetra-acetylated on lysines 5, 8, 12 and 16 (hereafter denoted H4ac4). A pull-down assay confirmed H4ac4 peptide recognition by both BDs, which was abolished by the YF mutation (Fig. 1f). Interestingly, *Ca*Bdf1 BD1 and BD2 bound tetra-acetylated H4 peptides with comparable strength (Fig. 1e), in contrast with mammalian Brd2, Brd4 and Brdt, which bind tetra-acetylated H4 more tightly through BD1 than through BD2 (refs 13,14), highlighting a certain redundancy in the ligand-binding activity of the two *Ca*Bdf1 BDs.

### Bdf1 BD functionality is essential in *C. albicans*.
We next asked whether Bdf1 BD function is important for the viability of *C. albicans*. Although a heterozygous *BDF1* deletion mutant generated in strain SN152 (derived from SC5314) exhibited no significant phenotype, we were unable to obtain a homozygous *bdf1Δ/Δ* mutant, suggesting that *BDF1* is essential. To confirm essentiality we placed the remaining allele of the *BDF1* gene in the heterozygous strain under the control of a conditional promoter and evaluated survival under repressive conditions. We used either a methionine-repressible promoter or a newly engineered tetracycline (Tet)-regulatable cassette compatible with animal studies. Tet-dependent gene expression in *C. albicans* is usually achieved by integrating a chimeric transactivator protein and a Tet-responsive promoter independently into the genome[33,34]. Here we constructed a cassette allowing integration of all required components in a single step. The cassette contains the transactivator (TetR-VP16), a selective marker (*ARG4*) and seven tandem Tet-operator elements, which we inserted upstream of the *BDF1* open reading frame (ORF) to generate strain *Δbdf1/pTetO-BDF1* (Fig. 2a). Immunoblotting with a polyclonal antibody developed in this study to allow specific *Ca*Bdf1 detection (Supplementary Fig. 1c,d) confirmed that Bdf1 was expressed from the *pTetO* promoter in the absence of doxycycline (Dox), albeit at a weaker level than from the endogenous *BDF1* promoter, and was effectively repressed in the presence of Dox (Fig. 2b). Strikingly, the growth of strain *Δbdf1/pTetO-BDF1* mirrored these expression levels: compared to wild type (WT), growth was reduced in the absence of Dox and abrogated in its presence (Fig. 2c). The phenotype was rescued by re-introducing a functional copy of *BDF1* (strain *BDF1-R/pTetO-BDF1*), confirming that *BDF1* is essential in *C. albicans*.

To verify the importance of BD function for fungal growth, we generated strains in which one or both Bdf1 BDs were inactivated by domain deletion or by the YF point mutation while the other

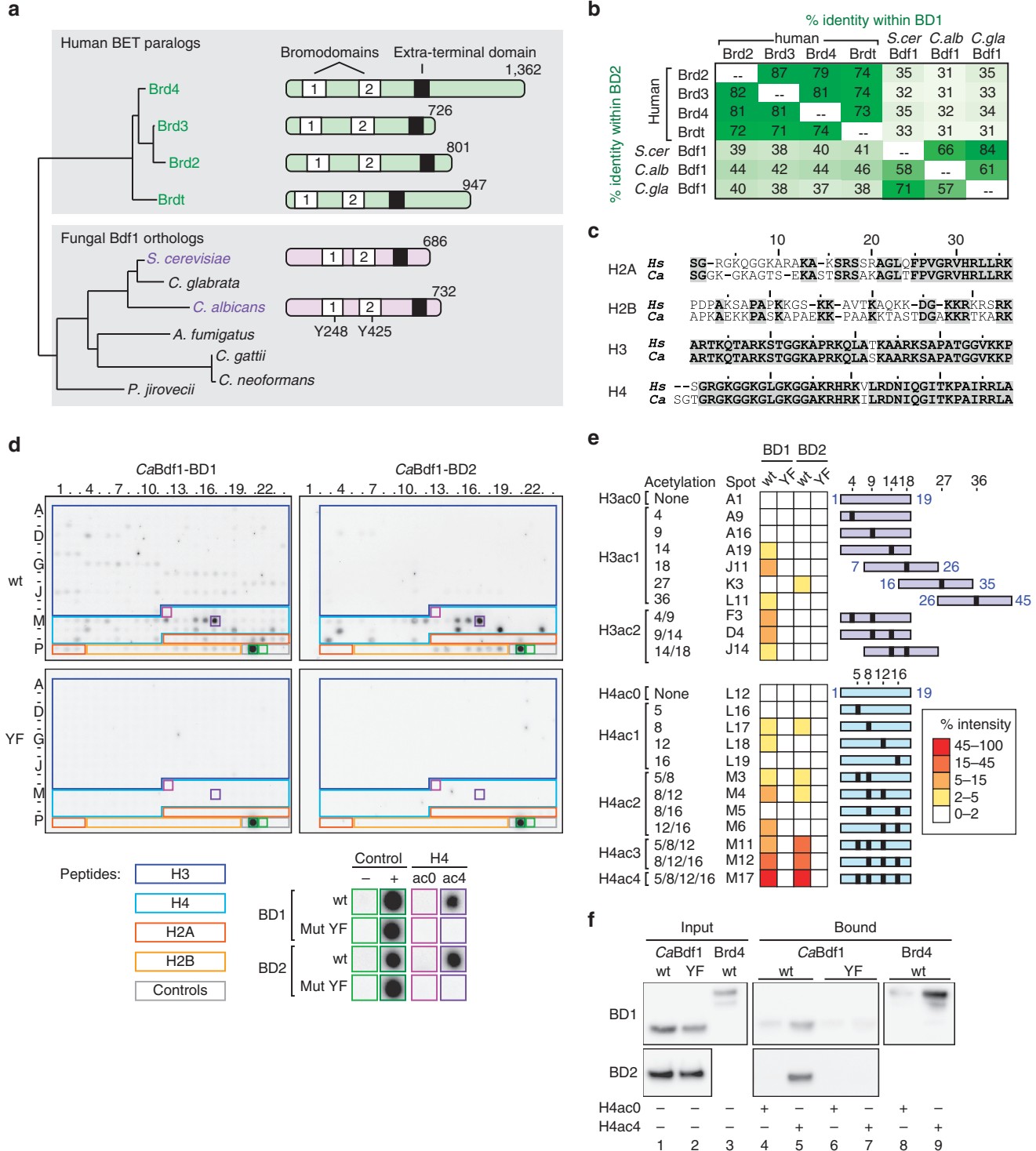

**Figure 1 | CaBdf1 BDs bind multi-acetylated histone tails.** (a) Phylogeny and domain organization of fungal Bdf1 and human BET proteins. The neighbour-joining tree based on an alignment of BET protein sequences was generated using programs ClustalX and NJPlot. (b) Sequence identity among human and fungal BET BDs. (c) Human (*Hs*) and *C. albicans* (*Ca*) histone tail sequences. (d) Binding of WT and mutant (YF) CaBdf1 BDs to a histone peptide microarray. The array comprised control peptides (grey outline) or N-terminal peptides from histones H3, H4, H2A and H2B (blue, cyan, dark orange and light orange outlines, respectively). Signals for background and positive-binding controls (boxed in light and dark green) or for H4ac0 and H4ac4 peptides (boxed in magenta and purple) are shown magnified below the array. Microarray data are summarized in Supplementary Data 1. (e) Binding intensities for H3 and H4 peptides. Peptides are shown as blue (H3) and cyan (H4) rectangles, with acetylation marks indicated by black boxes. (f) Pull-down assay. Immobilized H4ac0 and H4ac4 peptides were incubated with GST-tagged CaBdf1 BDs or with the corresponding YF mutants. After washing, bound proteins were eluted and visualized by western blotting with an anti-GST antibody. The full blots are shown in Supplementary Fig. 13a.

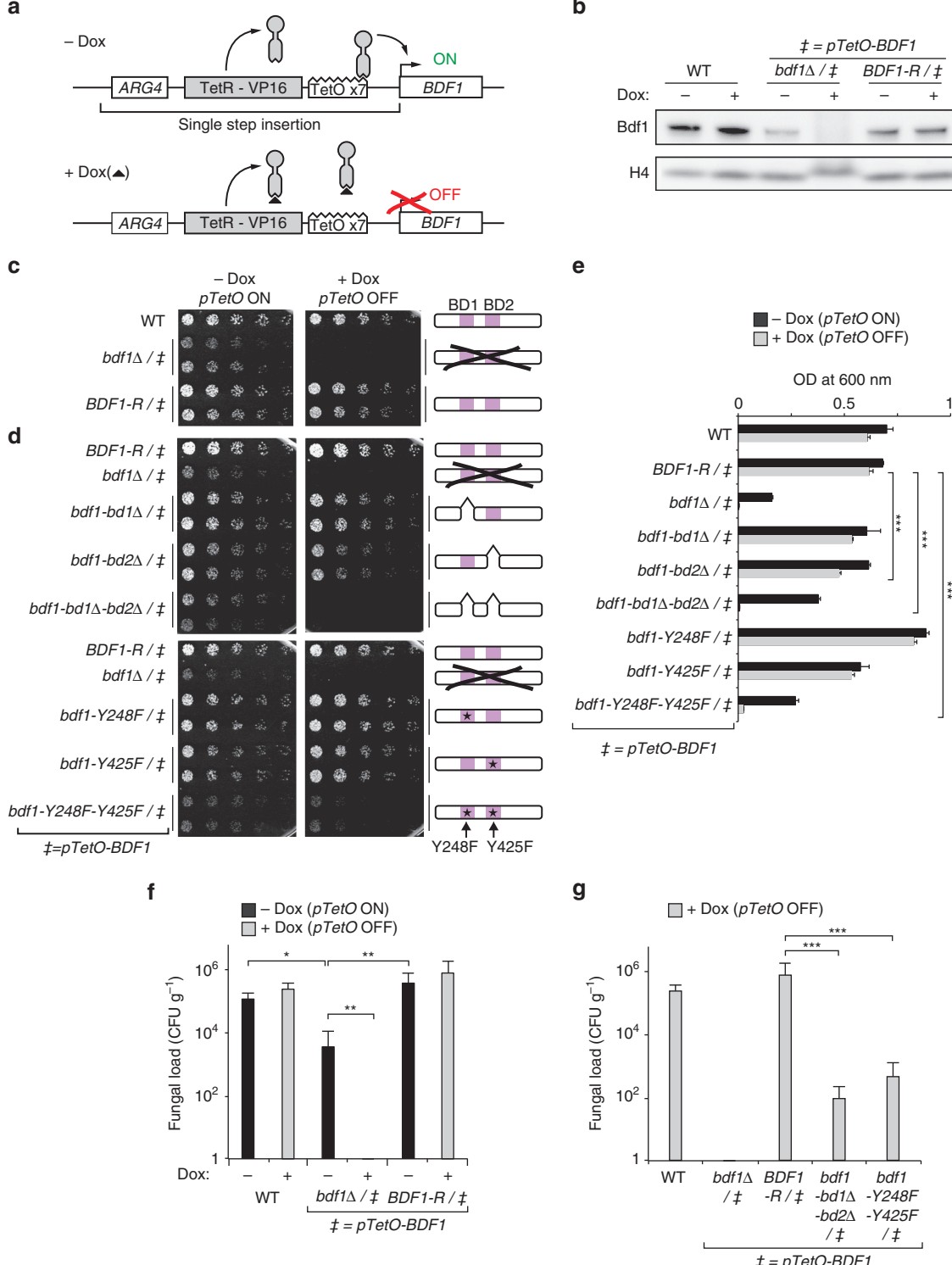

**Figure 2 | Bdf1 BDs are essential for *C. albicans* viability and virulence.** (**a**) Tet-OFF construct used in this study. Dox inhibits the binding of TetR-VP16 to the TetO, preventing *BDF1* transcription. (**b**) Bdf1 protein expression in different strains. The full blots are shown in Supplementary Fig. 13b. (**c,d**) Colony formation assays showing the effect of (**c**) Bdf1 repression and (**d**) Bdf1 BD inactivation on growth. This experiment was repeated three times with similar results. (**e**) Growth assays in liquid media. An equal fungal load was seeded for each strain and growth monitored by optical density at 600 nm. Mean and s.d. values are shown from three independent experiments. \*\*\**P* ≤ 0.01. *P* values were determined in a two-sided Welch *t*-test. (**f**) Kidney fungal load of mice injected with strains shown in (**c**) measured 7 days post infection, showing that Bdf1 is required for virulence. Data shown are mean and s.d. values (*n* = 6). \**P* ≤ 0.07; \*\**P* ≤ 0.05. *P* values were determined using a two-sided Wilcoxon rank sum test with continuity correction. (**g**) Kidney fungal load of mice infected with strains shown in (**d**), showing the loss of virulence on BD inactivation. Data shown are mean and s.d. values (*n* = 6). \*\*\**P* ≤ 0.01; *P* values were determined using a two-sided Wilcoxon rank sum test with continuity correction.

WT allele is expressed from the Dox-repressible promoter. Strains in which both BDs were inactivated grew as poorly as the conditional deletion mutant, whereas strains in which only BD1 or BD2 was inactivated displayed milder growth defects, with BD2 inactivation yielding the more pronounced defect (Fig. 2d). Additional assays evaluating stress resistance or cell wall integrity did not reveal any significant phenotype. Growth rates in liquid media recapitulated the phenotypes observed in the colony formation assay (Fig. 2e). Analogous results were obtained when *BDF1* expression was repressed using the methionine-regulatable promoter (Supplementary Fig. 2). Thus, *C. albicans* viability *in vitro* requires the presence of at least one functional BD within Bdf1.

**Bdf1 BDs are required for virulence in a mouse model.** Using our Tet-OFF system for Dox-repressible Bdf1 expression, we verified the role of Bdf1 in a murine model of invasive candidiasis. Initially, mice were injected with either WT *C. albicans*, the conditional *BDF1* deletion strain (*bdf1Δ/pTetO-BDF1*) or the rescue strain (*BDF1-R/pTetO-BDF1*). Mice injected with the WT or rescue strains exhibited a high fungal load ($>110,000\,CFU\,g^{-1}$) in the kidney 7 days post infection (Fig. 2f). The fungal load observed with strain *bdf1Δ/pTetO-BDF1* was reduced by 30-fold ($\sim 3,500\,CFU\,g^{-1}$) in the absence of Dox and was abolished in its presence, mirroring the growth phenotypes observed *in vitro* (Fig. 2c). Subsequently, we evaluated the virulence of strains in which both Bdf1 BDs were deleted or inactivated by the YF point mutations (Fig. 2g). These strains exhibited markedly reduced fungal loads (by $\sim 8,800$- and 1,600-fold, respectively), consistent with the *in vitro* growth rates associated with these mutations (Fig. 2d,e and Supplementary Fig. 2). Hence, Bdf1 BD functionality appears critical for the virulence of *C. albicans* in vivo.

**CaBdf1 BDs are resistant to human BET inhibitors.** For Bdf1 inhibition to be a feasible antifungal therapeutic strategy, small-molecule inhibitors need to discriminate fungal from human BET BDs. To probe the similarity between the ligand-binding pockets of *C. albicans* and human BET BDs, we asked whether human BET inhibitors (BETi) could inhibit *Ca*Bdf1 BDs. We used a homogeneous time-resolved fluorescence (HTRF) assay to evaluate the ability of four chemically diverse BETi compounds (JQ1, PFI-1, IBET-151 and bromosporine) to inhibit BD binding to an H4ac4 peptide (Fig. 3a). The four compounds inhibited human Brd4 BD1 and BD2 with median inhibitory concentrations (IC$_{50}$ values) in the low nM range (3–95 nM), consistent with previous reports[16,35,36]. In contrast, the *Ca*Bdf1 BDs were relatively insensitive to these inhibitors, with IC$_{50}$ values between 0.3 and $>10\,\mu M$, or approximately a two- to threefold log-reduction in sensitivity (Fig. 3b). These results were confirmed for JQ1 by isothermal titration calorimetry (ITC): whereas JQ1 bound Brd4 BD1 tightly ($K_d$ of 62 nM), no binding was detected for *Ca*Bdf1 BD1 or BD2 (Fig. 3c and Supplementary Table 1). None of the BETi compounds significantly inhibited growth of *C. albicans in vitro* at $10\,\mu M$ concentration (Fig. 3d). This finding is consistent with the poor IC$_{50}$ values observed for JQ1 and PFI-1 towards *Ca*Bdf1 BDs. It also indicates that IBET-151 and bromosporine, which display (sub)micromolar IC$_{50}$ values, have poor cellular potency, possibly due to mechanisms known to reduce drug potency in *C. albicans*, including cell wall and plasma membrane permeability barriers[37,38] and the activity of efflux pumps leading to rapid drug extrusion[39,40]. In summary, *Ca*Bdf1 and human BET BDs have distinct BETi-binding activity, supporting the feasibility of targeting *Ca*Bdf1 BDs with highly selective small-molecule inhibitors.

**BETi resistance by *Ca*Bdf1 is due to smaller pocket residues.** To identify features differentiating the ligand-binding pockets of human and fungal BET BDs, we determined crystal structures for *Ca*Bdf1 BD1 and BD2 (Supplementary Table 2). These structures exhibit the canonical BD fold (comprising helices Z,A,B and C, with the ZA and BC loops defining the ligand-binding pocket; Fig. 4a) and closely resemble those of their human BET counterparts (mean pairwise root-mean-square deviations (RMSDs) $\leq 1.5\,Å$; Supplementary Table 3). The ligand-binding loops are structurally well conserved, including nearly all water molecules implicated in ligand recognition (Supplementary Fig. 3), although the overall binding surfaces are less negatively charged in *Ca*Bdf1 (Fig. 4b). Superimposing the *Ca*Bdf1 BD structures onto that of JQ1-bound Brd4 BD1 reveals no major steric clashes between JQ1 and the fungal BDs. However, whereas JQ1 fits snugly into the Brd4-binding pocket, the fit with the *Ca*Bdf1-binding pockets is poor. In Brd4 the fused ring system of JQ1 is sandwiched between the 'WPF' shelf (Trp81, Pro82 and Phe83) and Leu residues 92 and 94, with the thiophene ring positioned within the 'ZA channel' defined by Trp81 and Leu92 (ref. 41). Replacement of Trp81 by a Val or Phe residue in *Ca*Bdf1 BD1 and BD2, respectively, widens the ZA channel, resulting in a poorer fit (black diamonds in Fig. 4c). Replacement of Brd4 residue Leu94 by a Val or Ile residue further contributes to the poor fit. In addition, the *p*-chlorophenyl group of JQ1 occupies a groove in Brd4, which is absent (BD1) or shallower (BD2) in *Ca*Bdf1 (asterisks in Fig. 4c). A similar lack of complementarity between the ligand and binding pocket explains the insensitivity of *Ca*Bdf1 BDs towards PFI-1, IBET-151 and bromosporine.

More generally, recognition of the four BETi compounds by human BET BDs involves 16 residues, including the eight signature residues previously used to classify human BD-binding sites[42] (Fig. 5). Five of these residues in BD1 and four in BD2 are invariant across human BET proteins but diverge in *Ca*Bdf1 (magenta arrows in Fig. 5), including the WPF-shelf Trp residue and the Leu residue opposite this shelf, at signature positions 1 and 3, respectively. Remarkably, nearly all of these residues have shorter side chains in *Ca*Bdf1 than in the human BET proteins, implying that selective *Ca*Bdf1 inhibition could conceivably be achieved via small molecules that clash sterically with the human, but not fungal, side chains. Interestingly, other pathogenic fungal Bdf1 sequences also diverge from human BET proteins at these residue positions, suggesting that selective chemical inhibition might also be feasible for these targets.

**CaBdf1 BDs can be selectively targeted by small molecules.** We next set out to identify small-molecule inhibitors that would target *C. albicans* Bdf1 BDs without inhibiting human BET BDs. We used our HTRF inhibition assay to screen a library of $\sim 80,000$ chemically diverse compounds, resulting in several hundred hits for each BD (Fig. 6a). Dose–response curves measured for *Ca*Bdf1 and Brd4 BDs identified 125 and 44 compounds selective for the fungal BD1 and BD2, respectively. In particular, several compounds possessing a dibenzothiazepinone scaffold exhibited low-micromolar IC$_{50}$ values towards *Ca*Bdf1 BD1 and no or only weak inhibition of Brd4 BD1 at the highest concentration ($20\,\mu M$) tested. One such inhibitor was compound **1** (Fig. 6b). ITC measurements showed that **1** binds *Ca*Bdf1 BD1 with a $K_d$ of $5\,\mu M$, consistent with the IC$_{50}$ value ($4.3\,\mu M$) determined by HTRF, whereas no or only modest binding and inhibition were detected for Brd4 BD1 and *Ca*Bdf1 BD2 (Fig. 6c,d and Supplementary Table 1). Furthermore, a BROMOscan screen from DiscoverX revealed that compound **1** showed no significant activity against 32 representative human (BET and non-BET) BDs (Supplementary Fig. 4); indeed, an ITC experiment

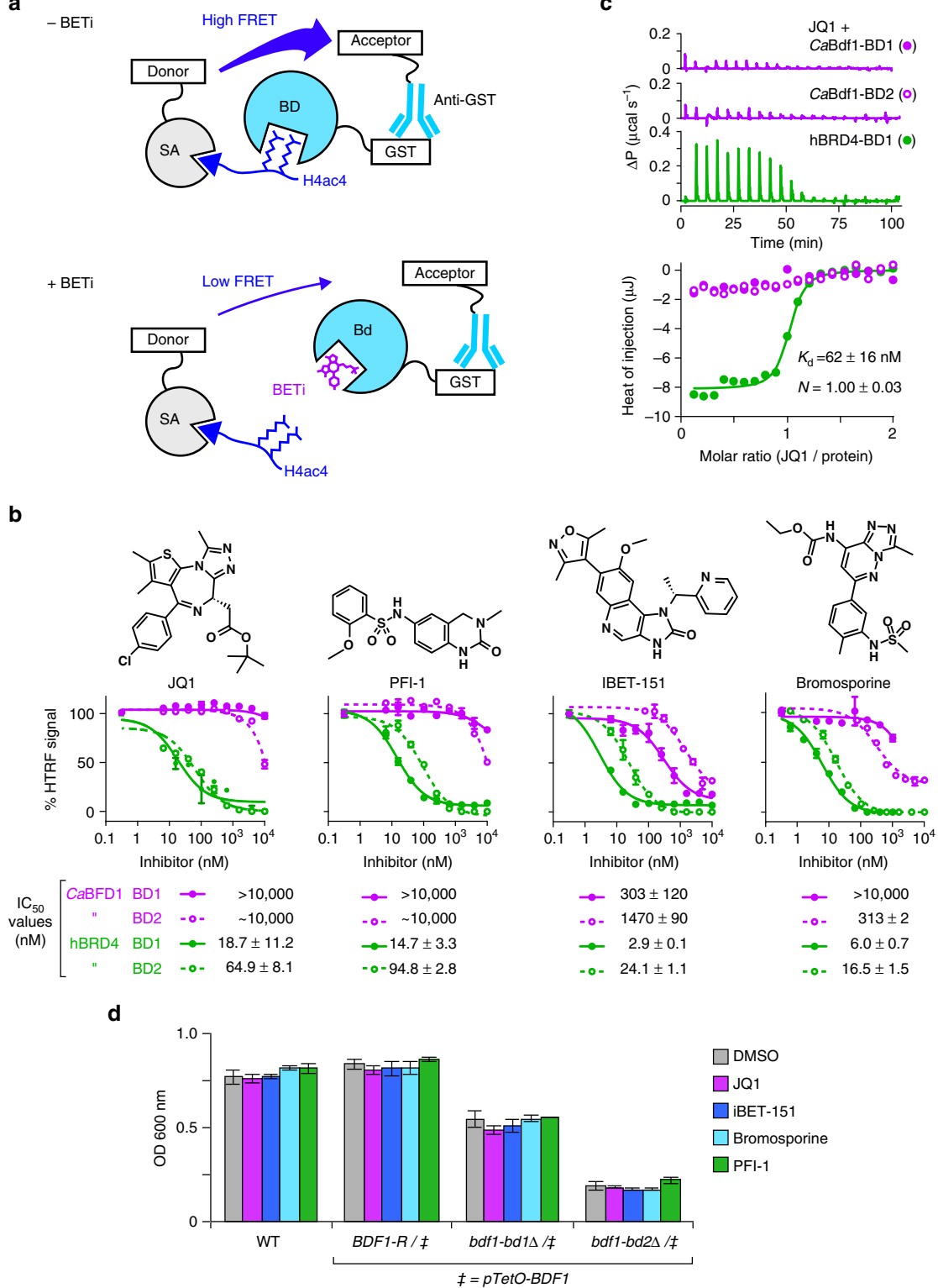

**Figure 3 | *Ca*Bdf1 BDs are resistant to BETi. (a)** HTRF assay. A biotinylated tetra-acetylated histone H4 peptide is bound to streptavidin beads coupled to the donor fluorophore. A GST-tagged BD is bound by an anti-GST antibody coupled to the acceptor fluorophor. Peptide binding by the BD results in FRET. The addition of a BETi reduces FRET. (**b**) HTRF assays performed on BDs from *Ca*Bdf1 and human Brd4 in the presence of the indicated BETi. Inhibition curves are shown as closed (BD1) and open (BD2) circles in green (Brd4) and magenta (Bdf1). $IC_{50}$ values are listed below each graph. Data represent the mean and s.d. values from three independent experiments. (**c**) Representative ITC experiments measuring the binding of JQ1 to *Ca*BDF1 BDs (magenta) and to human Brd4 BD1 (green). The values indicated for $K_d$ and $N$ represent the mean and s.d. from three independent experiments. See also Supplementary Table 1. (**d**) BET inhibitors do not affect *C. albicans* growth, even when Bdf1 BD1 or BD2 is deleted. Inhibitors were tested at 10 μM concentration. Experiments were performed in the presence of doxycycline to repress expression from the *pTetO*-BDF1 allele. Data represent the mean and s.d. values from three independent experiments.

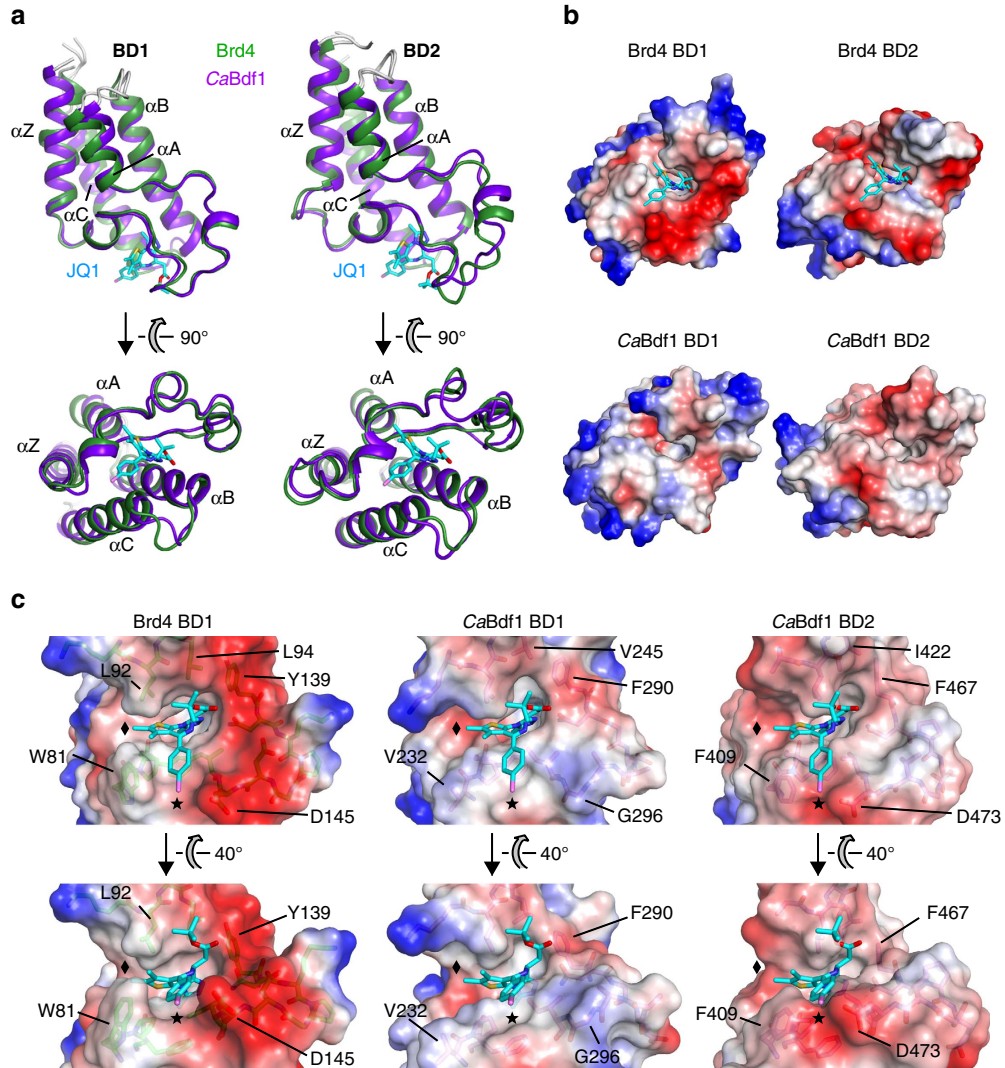

**Figure 4 | Structural basis of BETi resistance by *Ca*Bdf1 BD1 and BD2.** (**a**) Alignment of BD1 (left) and BD2 (right) structures from human Brd4 (green; PDB codes 3MXF and 2OUO) and *Ca*Bdf1 (magenta). For clarity, the 18 and 20 N-terminal residues preceding helix αZ are omitted from the Brd4 BD1 and *Ca*Bdf1 BD1 structures, respectively. Brd4-bound JQ1 is shown in cyan. (**b**) Plots of electrostatic surface potential of BD1 (left) and BD2 (right) structures from human Brd4 and *Ca*Bdf1. Regions of negative and positive potential are shown in red and blue, respectively. JQ1 bound to Brd4 is shown in cyan. The binding surfaces of the fungal BDs are less negatively charged compared to human BET BDs, suggesting differences in the binding partners of these domains. (**c**) Comparison of the ligand-binding pockets of *Ca*Bdf1 BD1 and BD2 with that of Brd4 BD1. JQ1 was superimposed on the *Ca*Bdf1 BDs via a structural alignment with Brd4 BD1. The diamond and asterisk indicate the ZA channel and surface groove which in Brd4 BD1 interact with the thieno and *p*-chlorophenyl groups of JQ1, respectively, and the corresponding positions in the *Ca*Bdf1 BDs. The view in the upper panels is that of (**b**) rotated 50° counter-clockwise.

confirmed that the two human BDs (SMARCA2 and SMARCA4) with the strongest BROMOscan response did not detectably bind compound **1** (Supplementary Fig. 5a). Compound **1** also exhibited low cytotoxicity towards HeLa and IMR90 (primary fibroblast) cells as measured in an MTT colorimetric assay ($EC_{50} \geq 100 \mu M$; Supplementary Fig. 6).

To understand the molecular basis of selectivity we solved the crystal structure of *Ca*Bdf1 BD1 bound to **1**. Compound **1** mimics the acetyllysine ligand by interacting with the conserved Tyr248 and Asn291 residues, forming direct and water-mediated hydrogen bonds via the thiazepinone carbonyl group (Fig. 6e–g). The dibenzothiazepinone ring system engages hydrophobic residues from the Val-Pro-Phe (VPF) shelf (Pro233), helix B (Phe290) and the ZZ′ loop (Val238, Leu243 and Val245), while the cyclopropyl group interacts with the VPF shelf (Pro233 and Phe234), helix B (Cys287) and the gatekeeper residue on helix C (Ile297). In contrast, the oxadiazole and pyrazine groups point

away from the pocket, adopting different orientations in the four complexes present in the crystal's asymmetric unit (Supplementary Fig. 7). Superimposing the ligand-bound structure with that of Brd4 BD1 results in a steric overlap between the larger Brd4 side chains at signature positions 1 and 3 (Trp81 and Leu94) and the oxadiazole and distal benzene groups of compound **1**, explaining why **1** fails to inhibit the human BD (Fig. 6f,h). Steric hindrance from the corresponding residues in *Ca*Bdf1 BD2 (Phe409 and Ile422) also account for this domain's insensitivity to compound **1**.

Similarly, we identified an imidazopyridine compound **2** (Fig. 7a), which inhibits *Ca*Bdf1 BD2 with high selectivity: HTRF and ITC assays yielded $IC_{50}$ and $K_d$ values of 1.1 and 2.1 μM, respectively, whereas human Brd4 BD2 and *Ca*Bdf1 BD1 were only weakly inhibited ($IC_{50} \geq 40 \mu M$) (Fig. 7b,c and Supplementary Table 1). Moreover, no significant inhibition of any human BDs was observed for compound **2** by BROMOscan

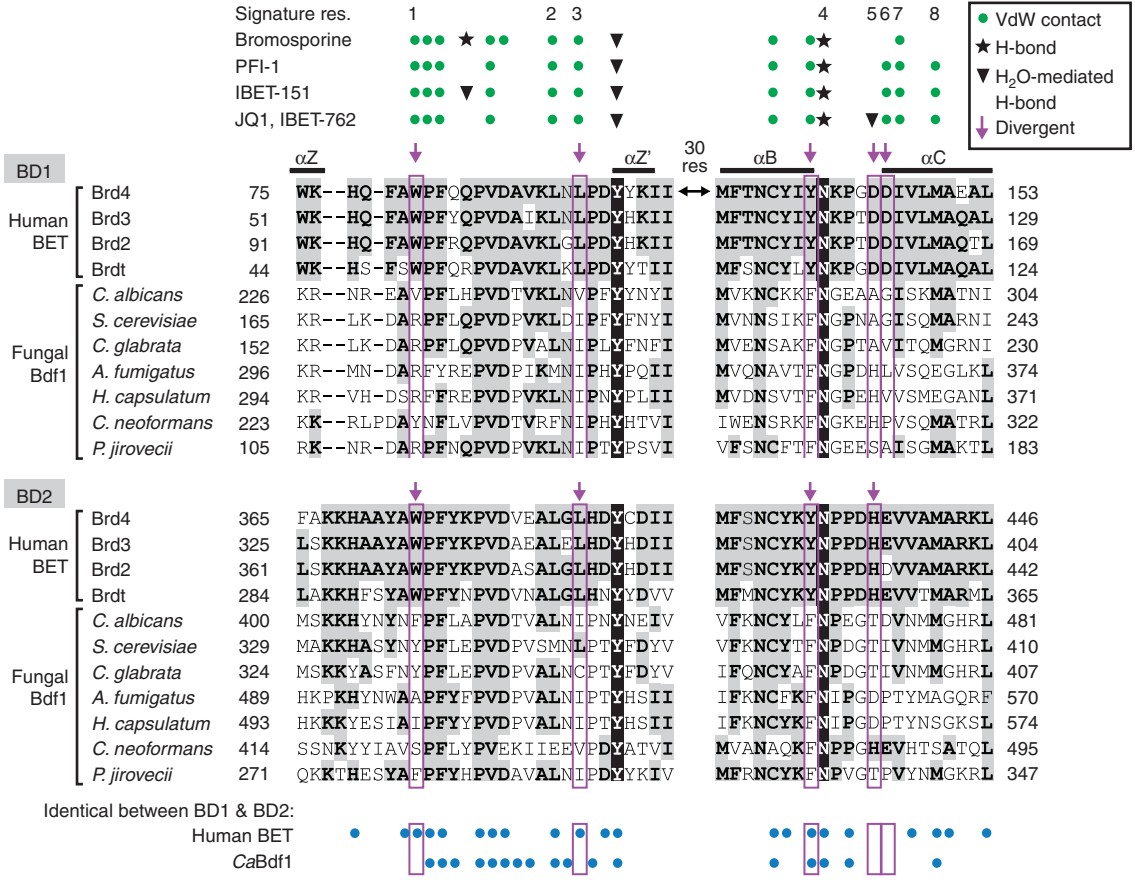

**Figure 5 | Structure-based sequence alignment of human and fungal BET proteins showing the ligand-binding region.** Residues highlighted in grey are conserved in at least three human BET sequences within each BD. The conserved Tyr and Asn residues involved in water-mediated and direct hydrogen bonds to the ligand are shown in inverse font. Human BET BD residues interacting with BETi compounds through direct or water-mediated hydrogen bonds or through van der Waals contacts are indicated by asterisks, arrowheads and green circles, respectively. Interactions are taken from the crystal structures of Brd4 BD1 bound to JQ1, IBET-762 or PFI-1 (PDB codes 3MXF, 3P5O and 4E96), from that of Brd2 BD1 bound to IBET-151 (PDB 4ALG), and from a structural alignment of Brd4 BD1 with human BRPF1 BD bound to bromosporine (PDB 4C7N). The signature residues used to classify human BD-binding sites are numbered 1–8 (ref. 42). Magenta arrows and boxes indicate BETi-contacting residues which are conserved in human BET proteins but differ in *Ca*Bdf1 and other fungal sequences. Blue dots below the alignment indicate residues which are identical between BD1 and BD2 across human BET proteins or within *Ca*Bdf1.

profiling and by an ITC assay on the two most sensitive BDs identified by this screen (SMARCA2 and SMARCA4; Supplementary Figs 4 and 5b); while only mild cytotoxicity was observed towards mammalian cells (EC$_{50}$ ∼60 μM; Supplementary Fig. 6). The crystal structure of *Ca*Bdf1 BD2 bound to **2** reveals that the compound forms a hydrogen bond via its amino group with the conserved Asn468 residue, as well as a cation–pi interaction involving the phenolic ring and the partial positive charge on Asn468 (Fig. 7d–f). However, unlike most BET inhibitors, a water-mediated hydrogen bond with Tyr425 is not observed for **2** (presumably because the ligand dislodges water molecule 1, consistent with the dislodgement of this water by a glycerol molecule in the unbound BD structure; Supplementary Fig. 3d). Instead, the compound's imidazolyl and hydroxyl groups form water-mediated hydrogen bonds with backbone atoms of the Phe-Pro-Phe (FPF) shelf (Pro410) and helix B (Val460). Replacement of the hydroxyl group in **2** by a hydrogen (**2a**) or fluorine (**2b**) atom severely reduces inhibitory activity, underscoring the importance of this group for high-affinity binding (Supplementary Fig. 8). The methylpyridinyl and phenolic groups are sandwiched between the FPF shelf and gatekeeper residue Val474 on one side, and residues Val415 and Leu420 on the other, while the tolyl group additionally contacts residues Ile422 and Phe467 (Fig. 7d–f). A structural alignment with Brd4 BD2

shows that larger Brd4 side chains at four of these contact positions, including signature positions 1,3 and 5 (Trp374, Leu387 and His437) and residue Tyr432 (Fig. 7e,g), reduces the volume available for compound **2**, explaining the inability of **2** to inhibit the human BD. In contrast, the poor affinity for *Ca*Bdf1 BD1 is due to smaller side chains at signature positions 1 and 3 (Val232 and Val245), resulting in poor complementarity between the binding pocket and ligand. The above results establish proof-of-principle that *Ca*Bdf1 BDs can be selectively targeted by small-molecule inhibitors *in vitro* without compromising the function of human BD-containing proteins.

**A Bdf1 BD1 inhibitor phenocopies genetic inactivation of BD1.** We next assessed the antifungal activity of compounds identified in our screen by evaluating their effect on *C. albicans* growth in liquid media. Not surprisingly, none of the compounds significantly inhibited growth of the WT strain, consistent with the fact that inactivating a single *Ca*Bdf1 BD only weakly affects growth (Fig. 2d,e) and that none of the compounds tested can individually inhibit both *Ca*Bdf1 BDs. Attempts to simultaneously inhibit both BDs by testing binary combinations of compounds also did not result in significant growth inhibition.

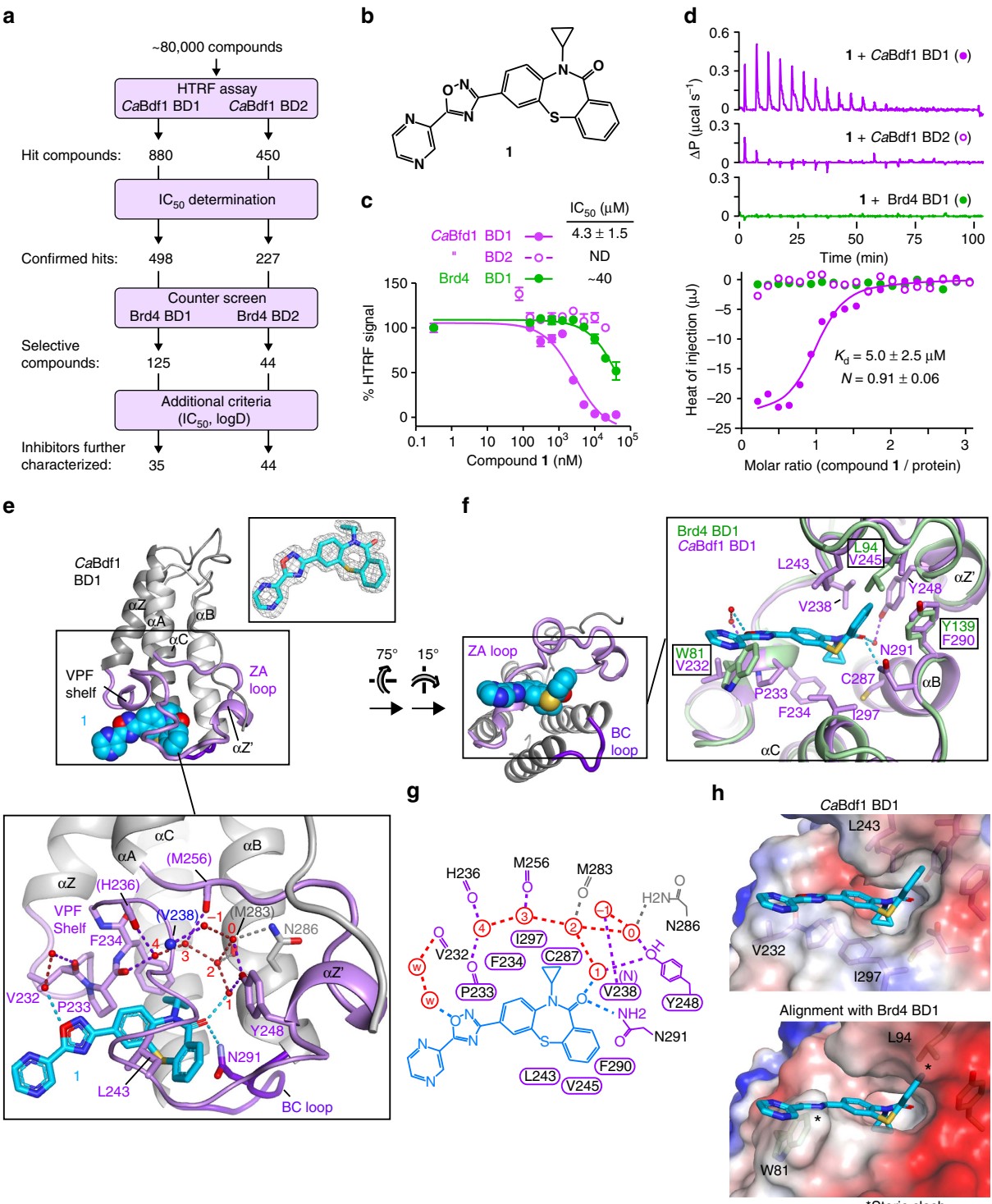

**Figure 6 | Identification of a selective CaBdf1 BD1 inhibitor.** (**a**) Screening strategy to identify selective CaBdf1 BD1 and BD2 inhibitors. Following the counter-screen step, most compounds yielded at least a 20-fold difference in IC₅₀ value relative to the human BD, although a few exhibited only a three- to fourfold difference. (**b**) Chemical structure of **1**. (**c**) HTRF assays showing selective inhibition of CaBdf1 BD1 by **1**. (**d**) Representative ITC experiments showing selective binding of **1** to CaBdf1 BD1. The values indicated for $K_d$ and $N$ represent the mean and s.d. from three independent experiments. See also Supplementary Table 1. (**e**) Crystal structure of CaBdf1 BD1 bound to **1**. Upper inset. Simulated-annealing omit $F_o$-$F_c$ density for **1** contoured at 3σ. Lower inset. Conserved water structure and hydrogen bonding interactions in the binding site. Residues interacting with **1** through direct and water-mediated hydrogen bonds (dashed lines) are shown in stick representation. Residues interacting through backbone atoms are labelled in parentheses. Water molecules are numbered as in ref. 70. (**f**) Alignment of CaBdf1 BD1 (violet) with Brd4 BD1 (green; PDB code 3UVW). Side chains are shown for CaBdf1 residues in contact with **1** and for the corresponding Brd4 residues if divergent from CaBdf1. (**g**) Schematic summary of ligand interactions. Hydrogen bonds are shown as dashed lines. Residues mediating van der Waals contacts with **1** are indicated by labels within a cartouche. Water molecules are indicated in red. (**h**) Surface representation showing the binding pocket of CaBdf1 BD1 bound to compound **1** (top) and that of Brd4 BD1 (bottom) superimposed on the fungal complex. Asterisks indicate steric overlap between compound **1** and Brd4 BD1 residues.

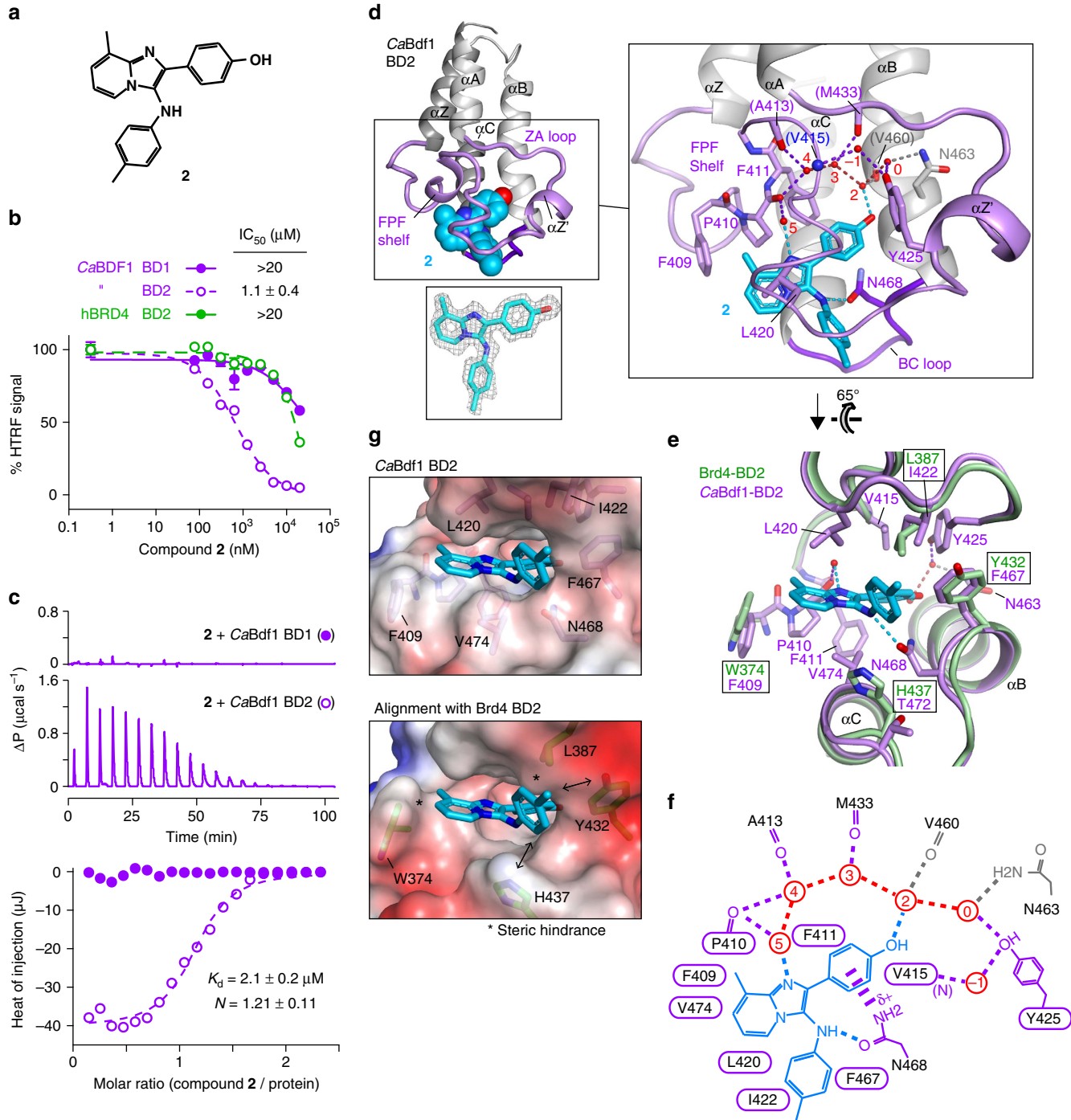

**Figure 7 | Identification of a selective CaBdf1 BD2 inhibitor. (a)** Chemical structure of compound **2**. (**b**) HTRF assays showing selective inhibition of CaBdf1 BD2 by **2**. (**c**) Binding of **2** to CaBdf1 BDs measured by ITC. The values indicated for $K_d$ and $N$ represent the mean and s.d. from three independent experiments. See also Supplementary Table 1. (**d**) Crystal structure of CaBdf1 BD2 bound to **2**. Lower inset. Simulated-annealing omit $F_o$-$F_c$ density for **2** contoured at $3\sigma$. Right inset. Details of the binding site. Residues interacting with **2** through direct and water-mediated hydrogen bonds (dashed lines) are shown in stick representation. Residues interacting through backbone atoms are labelled in parentheses. Water molecules are numbered as in ref. 70. (**e**) Alignment of CaBdf1 BD2 (violet) with Brd4 BD2 (green; PDB code 4Z93). Side chains are shown for CaBdf1 residues in contact with **2** and for the corresponding Brd4 residues if divergent from CaBdf1. (**f**) Schematic summary of interactions. Hydrogen bonds are shown as dashed lines. Water molecules are in red. The cation–pi type interaction between the partially charged N468 amino group and the phenol ring is shown as a thick dashed line. Residues mediating van der Waals contacts with **2** are indicated by labels within a cartouche. (**g**) Surface representation showing the binding pocket of CaBdf1 BD2 bound to compound **2** (top) and that of Brd4 BD2 (bottom) superimposed on the fungal complex. Asterisks indicate close contacts predicted to sterically inhibit the recognition of **2** by Brd4 BD2. Double arrows indicate short distances which may contribute additional steric hindrance.

We therefore tested individual compounds on mutant strains in which either BD1 or BD2 was inactivated. Most inhibitors, including compounds **1** and **2**, showed little antifungal activity against these strains. We speculate that the compounds failed to enter the fungal cell because of cellular permeability barriers[37,38], or were extruded by efflux pumps[39,40] or metabolized before a

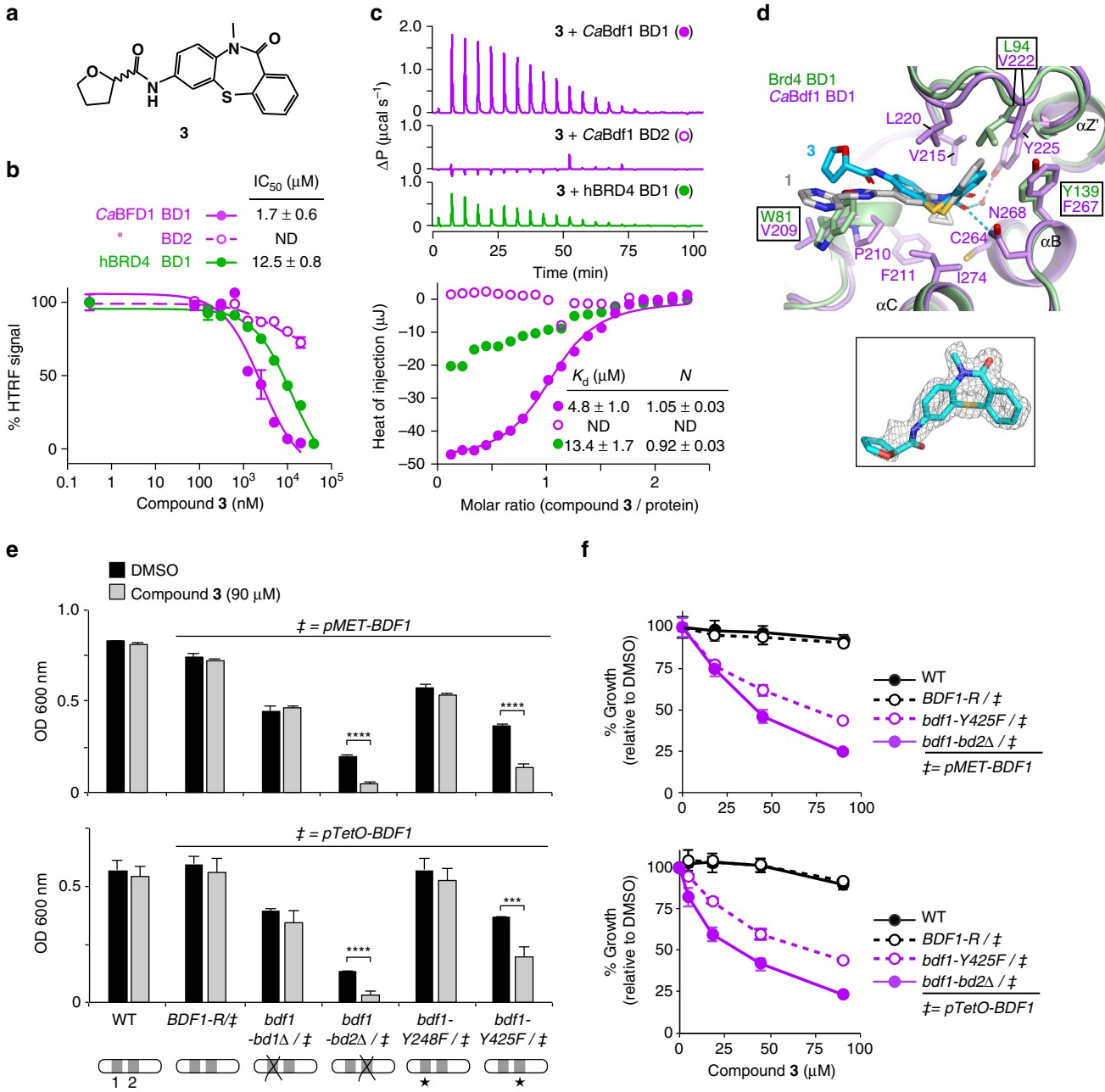

**Figure 8 | A *Ca*Bdf1 BD1 inhibitor phenocopies the effect of BD1 inactivation on *C. albicans* growth.** (**a**) Chemical structure of compound **3**. (**b**) HTRF assay showing that **3** inhibits *Ca*Bdf1 BD1 preferentially over *Ca*Bdf1 BD2 and with modest selectivity over Brd4 BD1. (**c**) Representative ITC experiments showing the binding of **3** towards the indicated BD. The values indicated for $K_d$ and $N$ represent the mean and s.d. from three independent experiments. See also Supplementary Table 1. (**d**) Crystal structure of *Ca*Bdf1 BD1 (violet) bound to **3** (*S* enantiomer) and alignment with CaBdf1 BD1 bound to **1** and with human Brd4 BD1 (green). Side chains are shown for *Ca*Bdf1 residues in contact with **3** and for the corresponding Brd4 residues if divergent from *Ca*Bdf1. Inset, simulated-annealing omit $F_o$-$F_c$ density for compound **3** contoured at 3σ. (**e**) Compound **3** compromises the *in vitro* growth of *C. albicans* when *Ca*Bdf1 BD2 is inactivated by deletion or point mutation. Strains were grown in the presence of methionine/cysteine or Dox to repress expression of the WT allele from the *pMET* (top) or *pTetO* (bottom) promoter, respectively. Data represent mean and s.d. values from three independent experiments. ***$P \leq 0.01$; ****$P \leq 10^{-3}$. $P$ values were determined in a two-sided Welch $t$-test. (**f**) Dose–response experiments showing the effect of **3** on Bdf1 mutant strains. Met/Cys or doxycyline were added to repress expression from the *pMET* (top) or *pTetO* (bottom) promoter, respectively. Data represent mean and s.d. values from three independent experiments.

significant growth defect was detected. The most active compound was dibenzothiazepinone **3**, an analogue of **1** in which the cyclopropyl, oxadiazole and pyrazine groups are replaced by smaller methyl, carboxyamidyl and tetrahydrofuran groups, respectively (Fig. 8a). HTRF and ITC assays showed that **3** inhibits *Ca*Bdf1 BD1 with low-micromolar affinity and three- to fivefold selectivity relative to human Brd4 BD1 (Fig. 8b,c and

Supplementary Table 1). The crystal structure of *Ca*Bdf1 BD1 bound to **3** shows that the compound is slightly rotated within the binding pocket compared to **1** (Fig. 8d). Consequently, in a structural alignment with Brd4 BD1, the steric hindrance observed for **1** due to the larger Trp81 and Leu94 side chains is largely relieved for compound **3**, explaining the reduced selectivity of this compound. (Both enantiomers of **3** are

equally active, consistent with the lack of interaction between the BD and the tetrahydrofuran ring; Supplementary Fig. 9). Liquid culture assays show that **3** has little effect on the growth of *C. albicans* strains expressing either WT Bdf1 or Bdf1 inactivated in BD1, but significantly inhibited the growth of strains inactivated in BD2 (Fig. 8e), consistent with the specificity of compound **3** for BD1. A dose–response curve revealed a cellular $EC_{50}$ value of $35 \pm 7 \mu M$, corresponding to a 20-fold cell shift relative to the $IC_{50}$ value determined by HTRF *in vitro* (or sevenfold relative to the $K_d$ by ITC) (Fig. 8f). At this concentration, compound **3** showed relatively little cytotoxicity towards mammalian cells ($EC_{50} > 100 \mu M$; Supplementary Fig. 6). Taken together, these results demonstrate the feasibility of using a small-molecule inhibitor to antagonize *Ca*Bdf1 function in the fungal cell.

## Discussion

We have shown that *C. albicans BDF1* deletion is lethal and that mutations abolishing BD-mediated ligand-binding activity severely compromise growth (Fig. 2c–e), validating BET BD inhibition as a potential antifungal strategy in *C. albicans*. Whereas the combined inactivation of both *Ca*Bdf1 BDs is lethal, that of a single BD leads to only a partial reduction in growth (Fig. 2d,e and Fig. 8e). This likely reflects functional redundancy between the two BDs, consistent with their similar binding profiles towards acetylated H4 tails (Fig. 1d,e). Growth inhibition appears slightly more pronounced on BD2 inactivation compared to that of BD1 (Fig. 2d,e). This contrasts with the finding that the selective inhibition of human BET BD2 in liver cancer cells induced only minor effects on transcriptional regulation[43] and that BD1 plays a more important role than BD2 in Brdt-mediated chromatin remodelling[44], in recruiting Brd3 to acetylated sites on GATA1 (ref. 45), and in chromatin binding by Brd4 (ref. 46). This discrepancy may partly reflect differences between human and *C. albicans* BET BD selectivity towards acetylated histone peptides, as well as different histone acetylation patterns in these species.

Because *C. albicans* strains inactivated in only one Bdf1 BD remain viable, a chemotherapeutic strategy targeting *Ca*Bdf1 would likely require the inhibition of both BDs to be effective. Such inhibition could be achieved via a single compound, which targets both BDs or through the combined use of two compounds, supplied independently or covalently linked as a bivalent BD inhibitor. Indeed, dual-warhead BET inhibitors that simultaneously engage both BDs within a single BET protein have recently been described, which possess greatly enhanced biochemical and cellular potency as well as increased efficacy in animal disease models relative to monovalent inhibitors[47–49]. Alternatively, one could envisage a Proteolysis Targeting Chimera approach, whereby a single BD inhibitor conjugated to a ligand for an E3 ubiquitin ligase would trigger the degradation of *Ca*Bdf1, as recently demonstrated for human BET proteins[50–53].

Compounds **1** and **2** target *Ca*Bdf1 BD1 and BD2, respectively, with low-micromolar affinity, yet do not significantly inhibit human BET or non-BET BDs, demonstrating that small-molecule inhibition of *Ca*Bdf1 BDs can be achieved with high selectivity. The selectivity of **1** and **2** for *Ca*Bdf1 results from steric incompatibility with bulkier side chains in the human BET BD-binding pockets, particularly at signature positions 1 and 3. Remarkably, all 169 *Ca*Bdf1-selective hits identified in our screen target either *Ca*Bdf1 BD1 or BD2, but not both, in contrast with many BETi compounds, which inhibit both BD1 and BD2 in human BET proteins. A likely explanation is that, whereas residues at signature positions 1 and 3 are conserved between human BET BD1 and BD2, they diverge between *Ca*Bdf1 BD1

and BD2, facilitating the identification of *Ca*Bdf1 inhibitors with single BD selectivity (Supplementary Fig. 10). Nevertheless, an inhibitor that targets both *Ca*Bdf1 BDs without inhibiting human BET BDs is in principle achievable, because the WPF-shelf Trp residue at signature position 3 in human BET BDs is bulkier than the corresponding (Val and Phe) residues in both *Ca*Bdf1 BD1 and BD2. For example, medicinal chemistry optimization of compound **2** could conceivably yield an analogue that would inhibit both *Ca*Bdf1 BD1 and BD2 without losing selectivity against human BET BDs.

Our chemical screen identified a dibenzothiazepinone compound, **3**, that selectively inhibited *Ca*Bdf1 BD1 and displayed antifungal activity against susceptible strains. Although we were unable to obtain biochemical evidence for a direct interaction between **3** and Bdf1 in *C. albicans* cells, the different susceptibility of Bdf1 mutant strains provides compelling evidence that Bdf1 BD1 is the intracellular target (Fig. 8e,f). Specifically, strains expressing Bdf1 mutants in which BD1 was the lone functional BD (mutants bdf1-bd2Δ and bdf1-Y425F) were sensitive to **3**, consistent with this compound's selectivity for *Ca*Bdf1 BD1, whereas all other strains with a functional BD2 were insensitive, arguing strongly against a potential off-target effect. Strikingly, compound **1** did not significantly inhibit growth, despite being an analogue of **3** with nearly identical binding affinity for *Ca*Bdf1 BD1. This makes the dibenzothiazepinone an attractive scaffold for structure–activity relationship studies, as these may reveal the molecular determinants of a compound's ability to reach its intracellular target and thereby lead to analogues with improved cellular potency.

In conclusion, Bdf1 BDs are required for the viability and virulence of *C. albicans* and can be selectively targeted by small-molecule inhibitors without compromising human BET BD function. These findings pave the way for the development of BET inhibitors as a novel class of antifungal therapeutics.

## Methods

**Chemicals.** BET inhibitors PFI-1, IBET-151 and bromosporine were purchased from Sigma while JQ1 was from Clinisciences (Nanterre, France). Compound **1** was purchased from ChemDiv as powder and dissolved in dimethylsulfoxide (DMSO) without further purification. Compounds **2** and **3** (the latter as the racemate), initially purchased from ChemDiv, were resynthesized in our laboratory where the individual enantiomers of **3** were also prepared (Supplementary Methods) and used for all *in vitro* and cell-based assays. Derivatives **2a** and **2b** were synthesized in our laboratory for this study (Supplementary Methods).

**Protein expression and purification.** *Proteins used for crystallization and for ITC.* DNA fragments encoding *Ca*Bdf1 BD1 (residues 193–327) or BD2 (residues 386–491) and human Brd4 BD1 (residues 22–204) were PCR amplified from genomic or cDNA libraries and cloned into a pETM11 vector as fusion constructs bearing an N-terminal His tag followed by a tobacco etch virus (TEV) protease cleavage site. (Primer sequences are listed in Supplementary Table 4.) Transformed *E. coli* strain BL21(DE3) (New England Biolabs, ref. C2527I) cells were grown in LB medium containing kanamycin (50 µg ml$^{-1}$) at 37 °C until reaching an $OD_{600}$ of 1, induced with 0.5 mM IPTG and further incubated at 16 °C for 12–20 h before collecting. Cells were resuspended in buffer A (50 mM Tris-HCl pH 7.5, 300 mM NaCl, 10% glycerol, 25 mM imidazole, 5 mM β-mercaptoethanol and protease inhibitors) and lysed by sonication. The cleared lysate was incubated with Ni-NTA resin (Qiagen) and washed with buffer A containing 0.5 M NaCl. Proteins were eluted with 250 mM imidazole, dialysed overnight in the presence of His-tagged TEV protease against buffer A containing no imidazole and incubated with Ni-NTA resin to remove His-tagged species. Proteins were further purified on a Superdex 75 10/300 (GE Healthcare) column in 50 mM Hepes pH 7.5, 150 mM NaCl, 0.5 mM dithiothreitol (DTT). Proteins were concentrated to $>20$ mg ml$^{-1}$ on a Centricon device (Millipore) and used for ITC, TSA and crystallization experiments. Human SMARCA2 and SMARCA4 BD plasmids bearing an N-terminal His tag and TEV protease site were purchased from Addgene (plasmid no. 73250 and 74664, respectively). Expression and purification were performed as above, except that buffer A was replaced by buffer B (50 mM Hepes pH 7.5, 500 mM NaCl, 5% glycerol, 5 mM imidazole and protease inhibitors) and gel filtration was performed in 10 mM Hepes pH 7.5, 500 mM NaCl and 5% glycerol. *Proteins used for HTRF assays.* GST-tagged human Brd4 BD2 (residues 349–460) was purchased from Reaction Biology Corp. Human Brd4 BD1 (residues

22–204) and CaBdf1 BD1 (residues 193–327) and BD2 (residues 361–501) were cloned into a pGEX4t1 vector as GST-tagged fusion proteins. Expression in E. coli strain BL21(DE3) cells was performed as for His-tagged constructs. Collected cells resuspended in 50 mM Tris-HCl pH 7.5, 150 mM NaCl and protease inhibitors were lysed by sonication. The clarified lysate was incubated with glutathione sepharose (GE Healthcare) and then washed with 50 mM Tris-HCl pH 7.5, 500 mM NaCl and 1% NP-40. Proteins were eluted with 10 mM glutathione and further purified on a Superdex 75 10/300 (GE Healthcare) column in 50 mM Hepes pH 7.5, 150 mM NaCl, 0.5 mM DTT.

**Pull-down and peptide array assays.** Biotinylated peptides corresponding to non-acetylated (H4ac0) and tetra-acetylated (H4K5acK8acK12acK16ac) histone H4 tails were synthesized by Covalab (Villeurbanne, France) and immobilized on Streptavidin-coated magnetic beads (Dynabeads MyOne Streptavidin C1; Thermo Fisher) according to the manufacturer's instructions. Beads were incubated with 1.25 µg of GST-tagged CaBdf1 BD1 or BD2 in binding buffer (50 mM Tris pH 7.4, 150 mM NaCl, 0.1%, NP-40, 10% glycerol 10%, 1 mM DTT) in a volume of 250 µl for 2 h at 4 °C and subsequently washed in binding buffer containing 500 mM NaCl. Bound proteins were eluted by boiling in SDS–PAGE sample loading buffer and analysed by western blot using an anti-GST antibody (GE Healthcare).

Histone peptide arrays were obtained from Active Motif (ref. 13005) and used in accordance with the supplier's recommendations. In short, arrays were blocked and incubated with 1 µM of purified CaBdf1 BD1 or BD2. Arrays were incubated with recombinant BDs for 2 h at 4 °C in binding buffer (Tris-HCl 50 mM pH 7.4, NP-40 0.1%, NaCl 150 mM, glycerol 10%). They were then washed in the same buffer and binding detected by a standard western blot procedure using an anti-GST antibody (Dutscher, ref. 27-4577-01). An antibody directed against a spotted c-myc peptide (supplied with the MODified Array Labeling Kit (Active Motif, ref. 13006) and used following the manufacturers' instructions at 1/2,000 dilution) was used as a positive control and its signal intensity was used for normalization. Assays were performed in duplicate. Images were acquired on a Bio-Rad ChemiDoc XRS Imaging machine and data were quantified using the array analyser software provided by Active Motif. The results of the quantification are summarized in Supplementary Data 1.

**ITC.** Calorimetric experiments were performed on a NanoITC calorimeter (TA Instruments) at 25 °C while stirring at 250 r.p.m. All proteins were buffer exchanged by gel filtration into 50 mM Hepes pH 7.5, 150 mM NaCl, 0.5 mM DTT, except for SMARCA2 and SMARCA4A BDs, which were exchanged into 10 mM Hepes pH 7.5, 500 mM NaCl and 5% glycerol. Typically, 10–100 µM compound and 60–900 µM protein were placed in the cell and syringe, respectively. Titrations consisted of an initial injection of 1.5 µl followed by 19 identical injections of 2.5 µl made at time intervals of 5 min. ITC data were corrected for the heat of dilution of injectant into buffer and analysed with software provided by the manufacturer using a single binding site model. The first data point was excluded from the analysis. Thermodynamic parameters determined from the ITC data are summarized in Supplementary Table 1.

**HTRF assay.** Biotinylated and non-biotinylated tetra-acetylated H4 peptides (H4K5acK8acK12acK16ac; denoted H4ac4) were synthetized by Covalab (Villeurbanne, France). HTRF reagents and buffers were purchased from Cisbio Bioassays and the assay performed according to the manufacturer's instructions. Specifically, the assay used a terbium(III) cryptate donor reagent conjugated to an anti-GST antibody (MAb anti-GST-Tb; ref. 61GSTTLA), a streptavidin-conjugated acceptor reagent (streptavidin-d2; ref. 610SADLA) and Cisbio proprietary buffers (EPIgeneous Binding Domain Diluent and Detection buffer; refs. 62DLBDDF and 62DB2FDG, respectively). Incubation with GST-tagged BDs and biotinylated H4ac4 brings the donor and acceptor into close proximity and allows FRET. The non-biotinylated H4ac4 peptide competes for binding and was used as a positive control for inhibition. GST-tagged proteins in 25 mM Hepes pH 7.5, 150 mM NaCl, 0.5 mM DTT were assayed at 5 nM final concentration. Biotinylated H4ac4 peptides were used at a final concentration of 50, 600, 300 or 400 nM in assays involving Brd4 BD1, Brd4 BD2, CaBdf1 BD1 and CaBdf1 BD2, respectively. The antibody-conjugated donor was used at 0.5 nM and the streptavidin-conjugated acceptor was used at 1/8 of the H4ac4 peptide concentration. Inhibitors were tested by performing a nine-point dilution series with a maximal final concentration of 10 µM (JQ1, IBET-151, PFI-1, bromosporine), 20 µM (**1**,**3**) or 40 µM (**2**). These concentrations allowed the DMSO concentration to remain below the limit (1% for CaBdf1 BD1 and Brd4 BD1, and 0.4% for CaBdf1 BD2 and Brd4 BD2) that allowed the assay to maintain a Z′ factor[54] ≥0.8. Components were incubated at room temperature for 1 h (CaBdf1 BD1 and Brd4 BD1) or at 4 °C for 6–24 h (CaBdf1 BD2 and Brd4 BD2). Experiments were performed in 384-well white plates (Greiner 781080) in a volume of 20 µl and analysed in a ClarioStar plate reader (BMG LABTECH). Excitation was at 330 nm and emission intensities were measured at 620 and 665 nm (corresponding to the donor and acceptor emission peaks, respectively; the 665/620 ratio is used to calculate the specific HTRF signal) with an integration delay of 60 µs and an integration time of 400 µs.

**High-throughput chemical screening.** The HTRF assay described above was miniaturized for performance in a 5 µl reaction volume in 1,536-well black plates. Approximately 80,000 compounds comprising the soluble diversity (ChemDiv), targeted diversity (ChemDiv) and 30 K diversity (LifeChem) collections were dispensed into wells by an Echo acoustic liquid dispenser. A master mix comprising MAb anti-GST-Tb donor, streptavidin-d2 acceptor, GST-tagged BD protein, biotinylated H4ac4 peptide was then added and the plates incubated for 6–24 h prior to reading. The primary screen was performed with compounds at a final concentration of 20 µM (CaBdf1 BD1) or 8 µM (CaBdf1 BD2), corresponding to a final DMSO concentration of 1% and 0.4%, respectively. Hits were initially confirmed by repeating the assay at a single concentration in triplicate, and subsequently by dose–response curves constructed using eight-point dilutions between 0 and 20 (or 8) µM.

**BROMOscan profiling.** BROMOscan profiling was performed by DiscoverX (Fremont, CA, USA). T7 phage strains displaying BDs were grown in parallel 24-well blocks in an E. coli host derived from the BL21 strain as described[55]. E. coli were grown to log-phase and infected with T7 phage from a frozen stock (multiplicity of infection = 0.4) and incubated with shaking at 32 °C until lysis (90–150 min). The lysates were centrifuged (5,000g) and filtered (0.2 µm) to remove cell debris. Streptavidin-coated magnetic beads were treated with biotinylated small molecule or acetylated peptide ligands for 30 min at room temperature to generate affinity resins for BD assays. Ligand-loading densities were optimized as described[56]. The liganded beads were blocked with excess biotin and washed with blocking buffer (SeaBlock (Pierce), 1% BSA, 0.05% Tween 20, 1 mM DTT) to remove unbound ligand and to reduce non-specific phage binding. Binding reactions were assembled by combining BDs, liganded affinity beads and test compounds in 1 × binding buffer (16% SeaBlock, 0.32 × PBS, 0.02% BSA, 0.04% Tween 20, 7.9 mM DTT). Test compounds were prepared as 1,000 × stocks in 100% DMSO and subsequently diluted 1:25 in monoethylene glycol. The compounds were then diluted directly into the assays such that the final concentrations of DMSO and monoethylene glycol were 0.1% and 2.4%, respectively. All reactions were performed in polypropylene 384-well plates in a final volume of 20 µl. The assay plates were incubated at room temperature with shaking for 1 h and the affinity beads were washed with wash buffer (1 × PBS, 0.05% Tween 20). The beads were then resuspended in elution buffer (1 × PBS, 0.05% Tween 20, 2 µM non-biotinylated affinity ligand) and incubated at room temperature with shaking for 30 min. The BD concentration in the eluates was determined by qPCR. Compounds **1** and **2** were screened at 10 µM concentration in duplicate. DMSO was used as a negative control and a small-molecule or peptide ligand specific for each BD in the screen was used as a positive control. Results are reported as percent control, where lower values indicate stronger inhibition. Percent control was calculated as $100 \times (t - p)/(n - p)$, where $t$ is the test compound signal and $p$ and $n$ are the positive and negative control signals, respectively.

**Crystallization and crystal structure determination.** Initial crystallization conditions were identified by the sitting drop vapour diffusion method at 4 °C using a Cartesian PixSys 4200 crystallization robot at the high-throughput crystallization laboratory of the EMBL Grenoble Outstation (https://htxlab.embl.fr). Crystals used for data collection were obtained by the hanging drop method at 4 °C by mixing 1 µl of the protein or protein/inhibitor sample with 1 µl of the reservoir solution, as follows. Unbound CaBdf1 BD1 (40 mg ml⁻¹) was mixed with 0.2 M ammonium iodide (pH 6.5) and 26% (w/v) PEG 3350. Unbound CaBdf1 BD2 (40 mg ml⁻¹) was mixed with 0.1 M Tris-HCl (pH 8.5) and 1 M ammonium phosphate. CaBdf1 BD1 bound to compound **1** was crystallized by mixing a solution of 20 mg ml⁻¹ protein and 0.4 mM inhibitor with 0.1 M Tris-HCl (pH 8.5), 23% (w/v) PEG MME 2000 and 10 mM NiCl₂. CaBdf1 BD1 bound to compound **3** was crystallized by mixing a solution of 25 mg ml⁻¹ protein and 1.5 mM inhibitor with 0.1 M sodium acetate (pH 4.6) and 2.4 M ammonium sulfate. CaBdf1 BD2 bound to compound **2** was crystallized by mixing a solution of 22 mg ml⁻¹ protein and 1.5 mM inhibitor with 0.1 M sodium acetate (pH 4.6) and 25% (w/v) PEG 3000. Crystals were flash-cooled in liquid nitrogen after being transferred into the well solution supplemented with the following cryo-protectants: unbound CaBdf1 BD1, unbound CaBdf1 BD2 and CaBdf1 BD1 bound to **1**, 30% (v/v) glycerol; CaBdf1 BD1 bound to **3**(S), 3.2 M ammonium sulfate; CaBdf1 BD2 bound to **2**, 25% (v/v) glycerol.

Diffraction data were collected at beamlines of the European Synchrotron Radiation Facility (ESRF), as indicated in Supplementary Table 2. Data collected on MarMOSAIC (ID23-2), Pilatus 6M (ID29) and Pilatus3 2M (ID30A-1) detectors were processed using X-ray Detector Software (XDS)[57] and programmes of the CCP4 suite[58]. Structures were solved by molecular replacement using Phaser[59]. Structures 2OSS and 2OUO (human Brd4 BD1 and BD2, respectively) were used as search models to solve the structures of CaBdf1 BD1 and BD2, respectively. Initial solutions were improved by automated protein chain tracing with ARP/WARP[60], followed by further manual building using Coot[61]. Inhibitor coordinates and .cif restraints files were generated using the Cactus online SMILES translator (https://cactus.nci.nih.gov/translate/) and JLigand[62], respectively. Structures were refined with Phenix[63] and validated using MolProbity[64] and the built-in tools in Coot. Examples of final $2F_o$-$F_c$ electron density are provided in Supplementary Figs 11 and 12. Data collection and refinement statistics are summarized in Supplementary Table 2.

**Cytotoxicity assays on human cells.** Proliferation of human cells was assessed using an MTT colorimetric assay (Cell Proliferation Kit I, Roche). HeLa (epithelial cells, ATCC number CCL-2) and IMR90 (primary fibroblasts cells, ATCC number CCL-186) cells were cultured in humidified atmosphere (37 °C and 5% $CO_2$) in DMEM medium containing 10% heat inactivated foetal calf serum and 2 mM glutamine. Cells were seeded at a concentration of 5,000 HeLa cells per well or 15,000 IMR90 cells per well in 100 µl culture medium containing the test compound (compounds **1,2,3**, amphotericin B or fluconazole) into 96 wells microplates (Falcon ref. 353072). Plates were incubated at 37 °C and 5% $CO_2$ for 28 h before adding 10 µl of MTT labelling reagent (final concentration 0.5 mg ml$^{-1}$) to each well. After incubating for a further 4 h, 100 µl of the solubilization solution were added in each well. Plates were allowed to stand overnight in the incubator before measuring the spectrophotometrical absorbance at 570 nm and at the reference wavelength of 690 nm in a ClarioStar plate reader. The values of $A_{570 nm} - A_{690 nm}$ were normalized relative to that obtained with vehicle (0.2% DMSO) and plotted against compound concentration.

**Generation of *C. albicans* mutant strains.** Plasmids used in this study to incorporate different mutations into the *BDF1* gene are listed in Supplementary Table 5. All DNA fragments were fused in a pCR2.1-TOPO vector using a Gibson assembly kit (New England Biolabs) and validated by sequencing. The *pMET3* promoter sequence was recycled from the plasmid pFA-ARG4-MET3p (ref. 65). The pTetO cassette (Fig. 2a) was generated by assembling the following fragments with the Gibson assembly kit (New England Biolabs) in a pCR2.1-TOPO vector: (i) *ARG4* gene from the plasmid pFA-ARG4-MET3p (ref. 65); (ii) a Tet-dependent-transactivation protein (TetR-VP16) obtained by gene synthesis with corrections for the genetic code of the CTU clade (Thermo), flanked by the promoter (500 bp) and terminator (500 bp) of the *C. albicans TDH3* gene; (iii) the repressible Tet-operator (pTetO) from the plasmid pCM184 (ref. 66). In this study, ~500 bp homologous regions were used to integrate the pTet cassette upstream of the *BDF1* ORF. The *BDF1* point mutant cassettes were obtained using the QuikChange site-directed kit (Agilent) with the *BDF1* plasmid pJG214 or pJG215. All cassettes were obtained by digesting 4 µg of the pCR2.1-TOPO plasmids and were transformed in *C. albicans* by a lithium acetate procedure, as previously described[67]. Mutations introduced in the *BDF1* gene in *C. albicans* were confirmed by PCR, sequencing of the mutated region in the genomic DNA and western blot. *C. albicans* strains used in this study are listed in Supplementary Table 6.

**Growth assays.** *Growth on solid media.* *C. albicans* strains were grown in SC media to an $OD_{600}$ of 0.5–0.8, pelleted and resuspended in sterile water at a final $OD_{600}$ of 0.13. Cells were spotted on solid media in a threefold dilution series starting at an $OD_{600}$ of 0.13. Plates were incubated at 30 °C for 1 day before imaging. Stress resistance was tested using the following conditions: $H_2O_2$ (4.5 and 6 mM), methylmethanesulfonate (0.005 and 0.02%), hydroxyurea (10 mM), nicotinamide (0.5, 1, and 4 mM), caffeine (15 mM), sorbitol (1, 1.5 and 2 M), Congo red (50 µg ml$^{-1}$), calcofluor white (20 µM), SDS (0.04%) and heat shock at 42 °C.

*Growth on liquid media.* *C. albicans* strains were grown at 30 °C in SC medium to an $OD_{600}$ of 0.5–0.8. For the evaluation of growth defects related to Bdf1 mutations, log-phase growing cells were counted using a Neubauer chamber and diluted in liquid media to a final concentration of 5,400 cells per ml per well in a sterile 96-well plate. Dox (50 µg ml$^{-1}$) or methionine (5 mM)/cysteine (0.25 mM) were added to the media if required. Plates were incubated at 30 °C and $OD_{600}$ was measured using a Multiskan FC Microplate Photometer (Thermo Fisher). For the evaluation of chemical compounds, strains were grown in SC medium supplemented with 50 µg ml$^{-1}$ Dox or 5 mM methionine/0.25 mM cysteine 24 h before counting and then seeded in 96-well plates at the same concentration as mentioned above, with or without chemical compounds. Statistical tests were performed using two-sided Welch *t*-tests.

**Analysis of whole-cell extracts and antibodies.** *C. albicans* strains were grown at 30 °C in SC media to an $OD_{600}$ of 0.5–0.8. Cells were lysed in FastprepTM (MPBiologicals) twice at 6.5 m s$^{-1}$ for 1 min with intermediate incubation on ice. Histone H4 antibody was purchased from Active Motif (ref. 39269) and the *Ca*Bdf1 antibody was developed in house using a full-length recombinant protein injected in rabbits (Covalab).

**Mouse experiments.** A mouse model of hematogenously disseminated candidiasis was used, as approved by the French National Animal Experimentation Ethics Committee (reference B385161006) under the file reference 2015042015538706 v4 (AFAPIS#644). *Candida* strains were inoculated from an overnight subculture in liquid yeast extract peptone dextrose (YPD). In each condition, six female BALB/c mice (8 weeks old) were injected in the lateral caudal vein with $2 \times 10^5$ cells of a *C. albicans* suspension. If necessary, Dox repression was added to the drinking water starting at 48 h before the injection until the day of killing. To limit severity and duration of pain, a pilot experiment established an end point 7 days post infection, when the virulence of each strain was assessed by kidney fungal burden evaluation, as described before[68,69]. Briefly, mice were killed, left kidneys were removed aseptically and weighed, and homogenized suspensions were cultured to evaluate colony-forming units (CFU). The final results were expressed as log CFUs per gram of tissue and statistically tested using two-sided Wilcoxon rank sum tests with continuity correction.

**Statistics.** Each statistical test is indicated in the legend of the corresponding figure and has been chosen for its compliance with the sample size. Sample size for mouse experiments was chosen as the best compromise between statistical power and regulations of the French National Animal Experimentation Ethic Committee. No inclusion/exclusion criteria were pre-established. Animal studies were neither randomized nor performed blind.

**Data availability.** Coordinates and structure factors have been deposited in the Protein Data Bank with accession codes 5N15 (unbound *Ca*Bdf1 BD1), 5N16 (*Ca*Bdf1 BD1 bound to **1**), 5N17 (*Ca*Bdf1 BD1 bound to **3**), 5N13 (unbound *Ca*Bdf1 BD2) and 5N18 (*Ca*Bdf1 BD2 bound to **2**). The authors declare that all other data supporting the findings of this study are available within the article and its Supplementary Information files, or from the corresponding authors on request.

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

## Acknowledgements

This work used the platforms of the Grenoble Instruct Center (ISBG: UMS 3518 CNRS-CEA-UJF-EMBL) with support from FRISBI (ANR-10-INSB-05-02) and GRAL (ANR-10-LABX-49-01) within the Grenoble Partnership for Structural Biology. We acknowledge the European Synchrotron Radiation Facility for provision of synchrotron radiation facilities and thank EMBL and ESRF staff for assistance at beamlines ID23-2, ID29 and ID30A-1, particularly M. Bowler and D. de Sanctis. We thank Myriam Ferro and Christophe Bruley for their general support, Sandrine Miesch-Fremy and Marie Arlotto for technical support, Joanna Timmins for access to and help with the CLARIOstar plate reader, Inah Kang for administrative support and EDyP team members for scientific discussions. This work was supported by grants from the FACE foundation (Partner University Fund to C.E.M. and C.P.), the National Institutes of Health (1R21AI113704 to C.E.M.), the Agence Nationale de Recherche (ANR-14-CE16-0027-01 (FungiBET) to C.P., J.G. and M.Co.; ANR-11-PDOC-011-01 (EpiGam) to J.G.; ANR-10-INBS-08 (ProFI) to J.G.; ANR-10-LABX-62-IBEID to C.D.), the EU FP7 Marie Curie Action (Career Integration Grant 304003 to J.G.) as well as by the USC Dornsife College of Letters, Arts and Sciences (E.F. and C.E.M.), a Chateaubriand Fellowship (E.F.) and a FINOVI fellowship from the Région Rhône Alpes, France (M.Cham.).

## Author contributions

F.M., C.E.M., J.G. and C.P. conceived and designed the study. F.M., E.F., M.Cham., N.Z., D.M., Y.Z., M.Ha., D.S., C.G., M.Cou., M.Chau., C.D., B.A.K., M.Hu., M.Cor., J.G. and C.P. performed experiments and analysed data. C.E.M., J.G. and C.P. wrote the manuscript. All authors discussed results and commented on the manuscript.

## Additional information

**Competing interests:** The authors declare no competing financial interests.

