## [Peer Review File · Nature Communications]

Reviewers' comments:

Reviewer #1 (Remarks to the Author):

The authors explored selective BET bromodomain inhibition as a potential antifungal target. The investigators used complementary genetic, pharmacologic, and structural approaches that support fungal Bdf1 as a valid and selective antifungal target. These observations are novel and should be of great interest to both the mycology community as well as the larger drug discovery community that is seeking to develop selective inhibitors for other disease states. In general, the studies provide convincing evidence of the authors claims. My only suggestion would be to incorporate mammalian controls in these studies to support the claims of selectivity and potential safety.

Reviewer #2 (Remarks to the Author):

Using genetics experiments and an animal model, Mietton et al. established that the BDF1 gene and the acetyllysine-binding function of the two bromodomains (BDs) of the Bdf1 protein are essential for *C. albicans* viability and virulence. This suggests that targeting its BDs by small molecules may lead to a new therapeutic strategy. Then, they developed novel, selective inhibitors of these *C. albicans* BDs. They solved the co-crystal structures of these BDs to reveal the structural basis for their selectivity. Finally, they showed that one of these compounds targeting the BD1 (compound 3) can inhibit the growth of the *C. albicans* strains that express Bdf1 with inactivated BD2. These results may be an important step toward a novel therapy targeting Bdf1, but several issues need to be addressed before the manuscript is accepted for publication.

Most importantly, the authors did not examine if the known and novel BD inhibitors actually inhibit the Bdf1 BDs in *C. albicans* cells. They claimed that inhibiting the Bdf1 BDs by small molecules can also inhibit the *C. albicans* growth. However, the correlation between the biochemical and growth inhibition activities is poor. Only compound 3 works, but IBET-151 and Bromosporine (with non-selective, higher *in vitro* activities against Bdf1), as well as compounds 1 and 2, do not in the growth inhibition assay. The authors simply speculated this is because of cell permeability and/or stability issues, without experimental support. Some experiments that monitor the Bdf1 inhibition activity of these compounds, such as Bdf1 chromatin immunoprecipitation, must be performed to support the authors' conclusion. It may also be beneficial to have an expression analysis, such as RNA-seq or microarray, to see if these compounds affect the expression of the Bdf1 target genes.

I wonder how selective compounds 1, 2, and 3 are. The HTRF assay shows that the IC₅₀ values of compound 1 against the *Candida* Bdf1 BD1 and human Brd4 BD1 are about 4 and 40 μM, respectively. The BROMOScan analysis shows that at 10 μM compound 1 hardly inhibits the human Brd4 BD1, while it has about 65% inhibition against the SMARCA2 and SMARCA4 BDs. It is difficult to compare the values from different experiments, but compound 1 may inhibit the *C. albicans* Bdf1 and human SMARCA proteins at similar levels. The BROMOScan analysis with *Candida* Bdf1, or the HTRF/ITC assay with purified human SMARCA BDs, will enable the direct comparison of the inhibition activity of compound 1 against these proteins and a better estimation of its specificity.

It may not be required, but will strengthen the authors' claim, if the effect of compound 3 on the *C. albicans* growth is shown using the mouse model.

Minor points

The structural work is valid and well presented. A supplementary figure may be added that shows

the structure of the human Brd4 BD1/2 bound with an acetylated histone peptide. This will help readers understand how BDs recognize acetyllysine, how the inhibitors occupy the acetyllysine binding site, and the role of the conserved Tyr whose Bdf1 counterparts are mutated in this study.

In Supplementary Table 2, the Rmerge, Rmeas, Rpim, and CC1/2 values are shown in percentages for the first four structures, but not for the BD2-compound 2 complex.

What is the difference between the top and bottom panels in Fig. 4f?

Reviewer #3 (Remarks to the Author):

Mietton and colleagues examine fungal bromodomains of the BET family as potential therapeutic targets in order to block fungal activity. This work is of particular importance as the emergence of drug-resistant fungal stains has called for new therapeutic approaches. Interestingly many fungal strains only have one copy of the BET gene Bdf1 which makes it a good candidate for targeting (based on the observation that deletion of both Bdf1 and Bdf2 in yeast is lethal). The authors establish in vitro the binding properties of Bdf1 BDs against histone peptides using SPOT arrays. Mutation of a conserved tyrosine in both BDs abolishes binding and pull down with WT vs mutant has the same effect. They show very nicely in a murine in vivo model that Bdf1 BD1/BD2 mutations can affect virulence, establishing an in vivo role for BD activity linking to invasive candidiasis. BETi targeting human BDs have no effect in vitro or in vivo when applied to Bdf1. The authors establish a structural rationale for this, by determining the crystal structures of Bdf1 BD1 and BD2 and finding the WPF topology is altered due to differences in primary sequence, resulting in a less ideal pocket for BETi binding. As fungal BD pockets are very different the authors suggest that it would be possible to get selective compounds that do not bind to human BDs. Indeed, HTRF screening of a library with 80k compounds yielded hits which were confirmed biophysically and did not show any activity against human BET BDs or other human BDs (tested by BromoScan). Compound 1 was found to bind BD1 and compound 2 was found to bind BD2. Both show low μM affinity. Structural characterization established the mode of binding and the differences to the human BRD4 BDs. Disappointingly, the lead compounds did not show any significant anti-fungal activity (ie growth inhibition) when used alone or in combination. This is unfortunate, since compound 3 which shows some degree of selectivity towards BD1, inhibits growth of BD2-deleted or BD2-mutated strains, suggesting that both BDs in Bdf1 are needed for proper protein function; it also suggests that compound 3 is cell permeable and not degraded. Overall this is a well designed, thorough and succinct study establishing the proof of principle for targeting Bdf1 BDs as a potential anti-fungal strategy.

Minor points

The authors incorrectly point out that CaBdf1 BD2 behaves as human BRD2 which only recognizes multiple acetylations via BD1 – BRD2 BD2 has been shown to bind 1:1 to both K5/K8 and K12/K16 by ITC.

Interestingly, strains carrying mutations or deletions of each of the two BDs showed growth defects which were more pronounced in the case of BD2 inactivation – this seems to be different from observations published before where selective inhibition of BD2 in human seems to have little effect in transcription for example – do the authors see a rationale in this?

The authors point out that all the hits identified in their screen hit wither BD1 or BD2 as opposed to human BETi which hit both domains. This is not an accurate statement – all human BETi referenced (JQ1, PFI1, IBET151, bromosporine) have low nM affinity against both BET BDs without showing a significant advantage against BD1 or BD2; however published compounds that show low μM activity and selectivity towards one of the two BDs exist, none of which has yielded a selective

low nM inhibitor, yet.

Although it would have been ideal to further pursue compound 3 and try to obtain a linked inhibitor hitting both BD1 and BD2, as exemplified with the recent disclosure of bivalent BET inhibitors (PMID: 27775715) the current study establishes a strong proof of principle for targeting fungal BDs without affecting human BET BDs, offering an attractive opportunity to develop novel antifungal agents. Can the authors comment?

Response to Reviewers

We are highly grateful to the reviewers for their constructive criticism and suggestions for improving the manuscript. Please find below our responses to their specific comments.

Reviewer #1:

The authors explored selective BET bromodomain inhibition as a potential antifungal target. The investigators used complementary genetic, pharmacologic, and structural approaches that support fungal Bdf1 as a valid and selective antifungal target. These observations are novel and should be of great interest to both the mycology community as well as the larger drug discovery community that is seeking to develop selective inhibitors for other disease states. In general, the studies provide convincing evidence of the authors claims. My only suggestion would be to incorporate mammalian controls in these studies to support the claims of selectivity and potential safety.

The revised manuscript now includes additional cytotoxicity assays on mammalian cells (**Supplementary Fig. 9**). These show that compounds **1-3** exhibit low cytotoxicity towards HeLa and primary fibroblast (IMR90) cells: EC₅₀ values (>100 μM for compounds **1** and **3** and >50 μM for compound **2**) are over 25-50 times higher than the corresponding IC₅₀ values observed *in vitro*. Importantly, relatively little cytotoxicity is observed on mammalian cells at concentrations of **3** that significantly inhibit the growth of the susceptible *C. albicans* strain.

Reviewer #2:

Using genetics experiments and an animal model, Mietton et al. established that the BDF1 gene and the acetyllysine-binding function of the two bromodomains (BDs) of the Bdf1 protein are essential for C. albicans viability and virulence. This suggests that targeting its BDs by small molecules may lead to a new therapeutic strategy. Then, they developed novel, selective inhibitors of these C. albicans BDs. They solved the co-crystal structures of these BDs to reveal the structural basis for their selectivity. Finally, they showed that one of these compounds targeting the BD1 (compound 3) can inhibit the growth of the C. albicans strains that express Bdf1 with inactivated BD2. These results may be an important step toward a novel therapy targeting Bdf1, but several issues need to be addressed before the manuscript is accepted for publication.

Most importantly, the authors did not examine if the known and novel BD inhibitors actually inhibit the Bdf1 BDs in C. albicans cells. They claimed that inhibiting the Bdf1 BDs by small molecules can also inhibit the C. albicans growth. However, the correlation between the biochemical and growth inhibition activities is poor. Only compound 3 works, but IBET-151 and Bromosporine (with non-selective, higher in vitro activities against Bdf1), as well as compounds 1 and 2, do not in the growth inhibition assay. The authors simply speculated this is because of cell permeability and/or stability issues, without experimental support.

As rightly noted by the Reviewer, none of the inhibitors investigated in this study except for compound **3** specifically inhibited the growth of the susceptible *C. albicans* strain (the strain expressing a Bdf1 mutant in which the lone functional BD was the BD specifically targeted by the inhibitor). The fact that Bdf1 BD-inactivating mutations compromise survival but that inhibitors which target these BDs in biochemical assays fail to inhibit growth shows that these compounds have low cellular potency. This is not surprising, given the efficient mechanisms known to reduce the cellular potency of drugs in *C. albicans*. These include fungal cell wall and plasma membrane permeability barriers and rapid drug extrusion by efflux pumps, which represent major challenges for antifungal drug development and the treatment of acquired resistance. Our revised manuscript now refers explicitly to these mechanisms. We also thank the Reviewer for pointing out that IBET-151 and bromosporine have higher *in vitro* activities against Bdf1 BDs relative to compounds **1-3**. Accordingly, we have clarified the following two statements in the Results section:

Original:

“None of the BETi compounds significantly inhibited growth of *C. albicans in vitro* (**Supplementary Fig. 3d**), although this might merely reflect inefficient entry into the fungal cell.”

Revised:

"None of the BETi compounds significantly inhibited growth of *C. albicans in vitro* at 10 μ M concentration (**Supplementary Fig. 3d**). This finding is consistent with the poor IC₅₀ values observed for JQ1 and PFI-1 towards CaBdf1 BDs. It also indicates that IBET-151 and bromosporine, which display (sub)micromolar IC₅₀ values, have poor cellular potency, possibly due to mechanisms known to reduce drug potency in *C. albicans*, including cell wall and plasma membrane permeability barriers^{37,38} and the activity of efflux pumps leading to rapid drug extrusion^{39,40}."

Original:

"Most inhibitors, including compounds **1** and **2**, showed little antifungal activity against these strains, presumably because they failed to enter the fungal cell or were metabolized or eliminated before a significant growth defect was detected."

Revised:

"Most inhibitors, including compounds **1** and **2**, showed little antifungal activity against these strains, presumably because they failed to enter the fungal cell (because of cellular permeability barriers^{37,38}) or were extruded by efflux pumps^{39,40} or metabolized before a significant growth defect was detected."

Some experiments that monitor the Bdf1 inhibition activity of these compounds, such as Bdf1 chromatin immunoprecipitation, must be performed to support the authors' conclusion. It may also be beneficial to have an expression analysis, such as RNA-seq or microarray, to see if these compounds affect the expression of the Bdf1 target genes.

We agree that additional data showing direct inhibition (or lack thereof) of Bdf1 BDs by BET inhibitors in *C. albicans* cells would enhance our manuscript. To this end, we have spent considerable effort over the last three months to obtain such data - regrettably, however, without success. In particular, we tried very hard to get the suggested Bdf1 ChIP experiment to work. Unfortunately, the anti-Bdf1 antibody developed in-house for this study, though efficient in Western blots, performs poorly at immunoprecipitating chromatin-bound Bdf1. Using established ChIP protocols previously optimized by one of the authors [CdE] for *C. albicans*, we tested many different experimental conditions (extracts, beads, antibody concentrations and ratios, washing conditions), as well as fresh antibodies immunopurified from a new batch of serum. Unfortunately, none of these efforts yielded reliable ChIP results. Ideally, we should now construct strains expressing the various forms of Bdf1 (WT and BD deletion and point mutants) bearing a FLAG- or HA-tag and then perform ChIP with an anti-FLAG or anti-HA antibody. We estimate that such an effort would necessitate an additional 4-5 months beyond the deadline for submitting the revised manuscript (note that genetic manipulations in *Candida albicans* are much slower and more laborious than in *S. cerevisiae* or *S. pombe*).

An additional complication is that Bdf1 target genes have not previously been characterized in *C. albicans*, rendering the analysis of BETi effects on gene expression nontrivial. While Bdf1 and Bdf2 are well characterized in *S. cerevisiae*, no specific functions or target genes have yet been established for Bdf1 in *C. albicans*. Based on homology with *S. cerevisiae* we tested a large selection of candidate ORFs by RT-qPCR; however, quantifying the expression of these genes did not yield significant differences between WT and Bdf1 mutant strains that could be specifically attributed to the Bdf1 mutation. A proper evaluation of the effect of BET inhibitors on gene expression would require a comprehensive RNA-seq analysis of experiments combining multiple strains (WT, bdf1-bd1YF, bdf1-bd2YF, bdf1-bd1YF-bd2YF) and conditions (\pm doxycyclin, \pm BET inhibitors), all performed in biological triplicate – an effort we feel is beyond the scope of the present study. Indeed, determining the function of Bdf1 in *C. albicans* and identifying its targets are the focus of a separate line of investigation in one of the authors' [JG] labs. This work, still in an early phase, is envisioned as an independent study to be published at a later date.

Despite being unable to obtain the requested biochemical data, we nevertheless feel that our evidence demonstrating an on-target effect for compound **3** is highly compelling. Our HTRF and ITC assays show that compound **3** specifically binds and inhibits CaBdf1 BD1, not BD2 (**Fig. 4b,c**). Accordingly, compound **3** only inhibits the growth of strains bearing BD1 as the lone functional BD; strains bearing a functional BD2 are

unaffected (**Fig 4e,f**). If inhibition were due to an off-target effect then one would expect the growth of strains expressing either WT Bdf1, the rescued form (BDF1-R), or the mutants inactivated in BD1 (*bdf1-bd1Δ* and *bdf1-Y248F*) also to be inhibited. However, this is not the case, and so we conclude that compound **3** inhibits growth of the susceptible strains by inhibiting *CaBdf1* BD1. To clarify and highlight the significance of these results we have modified the second last paragraph in the Discussion as follows:

Original:

“The dibenzothiazepinone **3** inhibited the growth of a *C. albicans* strain expressing a BD2-inactivated Bdf1 mutant, consistent with the compound’s selectivity for *CaBdf1* BD1, but not that of a strain expressing a BD1-inactivated mutant, arguing strongly against an off-target effect.”

Revised:

“Our chemical screen identified a dibenzothiazepinone compound, **3**, that selectively inhibited *CaBdf1* BD1 and displayed antifungal activity against susceptible strains. Although we were unable to obtain biochemical evidence for a direct interaction between **3** and Bdf1 in *C. albicans* cells, the different susceptibility of Bdf1 mutant strains provides compelling evidence that Bdf1 BD1 is the intracellular target (**Fig. 4e,f**). Specifically, strains expressing Bdf1 mutants in which BD1 was the lone functional BD (mutants *bdf1-bd2Δ* and *bdf1-Y245F*) were sensitive to **3**, consistent with this compound’s selectivity for *CaBdf1* BD1, whereas all other strains with a functional BD2 were insensitive, arguing strongly against a potential off-target effect.”

I wonder how selective compounds 1, 2, and 3 are. The HTRF assay shows that the IC₅₀ values of compound 1 against the Candida Bdf1 BD1 and human Brd4 BD1 are about 4 and 40 μM, respectively. The BROMOscan analysis shows that at 10 μM compound 1 hardly inhibits the human Brd4 BD1, while it has about 65% inhibition against the SMARCA2 and SMARCA4 BDs. It is difficult to compare the values from different experiments, but compound 1 may inhibit the C. albicans Bdf1 and human SMARCA proteins at similar levels. The BROMOscan analysis with Candida Bdf1, or the HTRF/ITC assay with purified human SMARCA BDs, will enable the direct comparison of the inhibition activity of compound 1 against these proteins and a better estimation of its specificity.

As suggested by the Reviewer, we performed additional ITC experiments with purified human SMARCA BDs. These experiments confirm that compounds **1** and **2** show no detectable binding to SMARCA2 or SMARCA4 BDs (**Supplementary Fig. 8**). Because the remaining 30 human BDs tested give even weaker BROMOscan signals in response to these compounds, the ITC results strongly support the conclusion that compounds **1** and **2** have little inhibitory activity towards human BDs. (Compound **3** was not tested because **Fig. 4** already shows that it is poorly selective, with an IC₅₀ for Brd4 BD1 only 7-fold higher than for *CaBdf1* BD1).

It may not be required, but will strengthen the authors' claim, if the effect of compound 3 on the C. albicans growth is shown using the mouse model.

Unfortunately, the low potency of compound **3** prevents us from performing the proposed experiment, which would require injecting mice with concentrations of compound **3** and of vehicle (DMSO) which are unacceptably high to receive the approval of the ethics committee on animal experimentation. We hope to perform such an experiment in the future once a more potent analog of **3** or other potent Bdf1 inhibitor is identified.

Minor points

The structural work is valid and well presented. A supplementary figure may be added that shows the structure of the human Brd4 BD1/2 bound with an acetylated histone peptide. This will help readers understand how BDs recognize acetyllysine, how the inhibitors occupy the acetyllysine binding site, and the role of the conserved Tyr whose Bdf1 counterparts are mutated in this study.

We have added the suggested figure as **Supplementary Fig. 1a**.

In Supplementary Table 2, the Rmerge, Rmeas, Rpim, and CC1/2 values are shown in percentages for the first four structures, but not for the BD2-compound 2 complex.

Thanks to the referee for spotting this. We have now corrected Supplementary Table 2.

What is the difference between the top and bottom panels in Fig. 4f?

These panels are analogous to the top and bottom panels of Fig. 4e, where the *BDF1* WT allele is expressed from either a pMET (top) or pTET(bottom) promoter. The following sentence has been added to the figure legend to make this clear:

"Met/Cys or doxycycline were added to repress expression from the pMET (top) or pTetO (bottom) promoter, respectively."

Reviewer #3:

Mietton and colleagues examine fungal bromodomains of the BET family as potential therapeutic targets in order to block fungal activity. This work is of particular importance as the emergence of drug-resistant fungal stains has called for new therapeutic approaches. Interestingly many fungal strains only have one copy of the BET gene Bdf1 which makes it a good candidate for targeting (based on the observation that deletion of both Bdf1 and Bdf2 in yeast is lethal). The authors establish in vitro the binding properties of Bdf1 BDs against histone peptides using SPOT arrays. Mutation of a conserved tyrosine in both BDs abolishes binding and pull down with WT vs mutant has the same effect. They show very nicely in a murine in vivo model that Bdf1 BD1/BD2 mutations can affect virulence, establishing an in vivo role for BD activity linking to invasive candidiasis. BETi targeting human BDs have no effect in vitro or in vivo when applied to Bdf1. The authors establish a structural rationale for this, by determining the crystal structures of Bdf1 BD1 and BD2 and finding the WPF topology is altered due to differences in primary sequence, resulting in a less ideal pocket for BETi binding. As fungal BD pockets are very different the authors suggest that it would be possible to get selective compounds that do not bind to human BDs. Indeed, HTRF screening of a library with 80k compounds yielded hits which were confirmed biophysically and did not show any activity against human BET BDs or other human BDs (tested by BromoScan). Compound 1 was found to bind BD1 and compound 2 was found to bind BD2. Both show low uM affinity. Structural characterization established the mode of binding and the differences to the human BRD4 BDs. Disappointingly, the lead compounds did not show any significant anti-fungal activity (ie growth inhibition) when used alone or in combination. This is unfortunate, since compound 3 which shows some degree of selectivity towards BD1, inhibits growth of BD2-deleted or BD2-mutated strains, suggesting that both BDs in Bdf1 are needed for proper protein function; it also suggests that compound 3 is cell permeable and not degraded. Overall this is a well designed, thorough and succinct study establishing the proof of principle for targeting Bdf1 BDs as a potential anti-fungal strategy.

Minor points

The authors incorrectly point out that CaBdf1 BD2 behaves as human BRD2 which only recognizes multiple acetylations via BD1 – BRD2 BD2 has been shown to bind 1:1 to both K5/K8 and K12/K16 by ITC.

We thank the Reviewer for pointing out this inaccuracy.

The statement referred to was initially worded in our manuscript as follows (Results, paragraph 1):

"For both BD1 and BD2 the strongest binding was observed with an H4 peptide tetra-acetylated on lysines 5, 8, 12 and 16 (hereafter denoted H4ac4). A pull-down assay confirmed H4ac4 peptide recognition by both BDs, which was abolished by the YF mutation (**Fig. 1f**). This finding differentiates CaBdf1 from human Brd2 and Brdt, which recognize multi-acetylated H4 only through BD1^{18,19}, highlighting a certain redundancy in the ligand-binding activity of the two CaBdf1 BDs."

We acknowledge that Brd2 and Brdt, as well as Brd4, can bind tetra-acetylated H4 through both BD1 and BD2. As shown in refs. 18 and 19, the affinity of BD1 is considerably (≥ 10 -fold) greater than that of BD2. We have modified the underlined sentence in the above statement to read as follows:

“Interestingly, *CaBdf1* BD1 and BD2 bound tetra-acetylated H4 peptides with comparable strength (Fig. 1e), in contrast with mammalian Brd2, Brd4 and Brdt proteins, which bind tetra-acetylated H4 more tightly through BD1 than through BD2^{13,14}, highlighting a certain redundancy in the ligand-binding activity of the two *CaBdf1* BDs.”

Interestingly, strains carrying mutations or deletions of each of the two BDs showed growth defects which were more pronounced in the case of BD2 inactivation – this seems to be different from observations published before where selective inhibition of BD2 in human seems to have little effect in transcription for example – do the authors see a rationale in this?

We thank the reviewer for this observation. We added the following to the Discussion to address this point:

“Growth inhibition appears slightly more pronounced upon BD2 inactivation compared to that of BD1 (Fig. 2d,e). This contrasts with the finding that the selective inhibition of human BET BD2 in liver cancer cells induced only minor effects on transcriptional regulation⁴³ and that BD1 plays a more important role than BD2 in Brdt-mediated chromatin remodelling⁴⁴, in recruiting Brd3 to acetylated sites on GATA1⁴⁵, and in chromatin binding by Brd4⁴⁶. This discrepancy may partly reflect differences between human and *C. albicans* BET BD selectivity towards acetylated histone peptides, as well as different histone acetylation patterns in these species.”

The authors point out that all the hits identified in their screen hit either BD1 or BD2 as opposed to human BETi which hit both domains. This is not an accurate statement – all human BETi referenced (JQ1, PF11, IBET151, bromosporine) have low nM affinity against both BET BDs without showing a significant advantage against BD1 or BD2; however published compounds that show low uM activity and selectivity towards one of the two BDs exist, none of which has yielded a selective low nM inhibitor, yet.

The Reviewer refers to the following statement in the Discussion:

"Remarkably, all 169 *CaBdf1*-selective hits identified in our screen target either *CaBdf1* BD1 or BD2, but not both, in contrast with most BETi compounds which inhibit both BD1 and BD2 in human BET proteins."

We did not intend to imply that there exist no human BETi compounds selective for only one BD, but rather that many human BETi compounds (those most commonly studied) exist which target both BDs. To clarify this we have changed "most BETi compounds" to "many BETi compounds" in the above sentence.

Although it would have been ideal to further pursue compound 3 and try to obtain a linked inhibitor hitting both BD1 and BD2, as exemplified with the recent disclosure of bivalent BET inhibitors (PMID: 27775715) the current study establishes a strong proof of principle for targeting fungal BDs without affecting human BET BDs, offering an attractive opportunity to develop novel antifungal agents. Can the authors comment?

We thank the referee for bringing attention to the newly published study. Indeed, since submitting our manuscript, several reports describing bivalent inhibitors have been published. To address this we have added the sentence underlined below to the Discussion:

“Because *C. albicans* strains inactivated in only one Bdf1 BD remain viable, a chemotherapeutic strategy targeting *CaBdf1* would likely require the inhibition of both BDs to be effective. Such inhibition could be achieved via a single compound which targets both BDs or through the combined use of two compounds, supplied independently or covalently linked as a bivalent BD inhibitor. Indeed, dual-warhead BET inhibitors that simultaneously engage both BDs within a single BET protein have recently been described which possess greatly enhanced biochemical and cellular potency as well as increased efficacy in animal disease models relative to monovalent inhibitors⁴⁷⁻⁴⁹.”

REVIEWERS' COMMENTS:

Reviewer #2 (Remarks to the Author):

Unfortunately, the authors were not able to provide direct evidence that compound 3 inhibits the Bdf1 function in cells, and thus did not properly address one of my previous concerns. The data in Fig. 4b,c are important, but I do not think they prove that compound 3's growth inhibition activity is via Bdf1. As compared with the WT and Bdf1-BD1-deficient strains, the Bdf1-BD2-deficient strains are already unhealthy without compound 3. Therefore, the possible effect of a synthetic growth defect with unknown off-target binding cannot be excluded.

I still think some data are required to show that compound 3 dissociates BD2-deficient Bdf1 from chromatin in cells. Such an experiment may be done in *C. albicans*, *S. cerevisiae*, or related organisms. It could be monitored by ChIP, biochemical fractionation, fluorescent imaging with a proper tag, or other methods. Transient expression systems may also be used.

The other issues were properly addressed in the revised manuscript. As a minor point, several lines seem to be missing in Supplementary Table 1.

Reviewer #3 (Remarks to the Author):

The authors have addressed all minor issues previously pointed out.

Response to Reviewers

Reviewer #2:

Unfortunately, the authors were not able to provide direct evidence that compound 3 inhibits the Bdf1 function in cells, and thus did not properly address one of my previous concerns. The data in Fig. 4b,c are important, but I do not think they prove that compound 3's growth inhibition activity is via Bdf1. As compared with the WT and Bdf1-BD1-deficient strains, the Bdf1-BD2-deficient strains are already unhealthy without compound 3. Therefore, the possible effect of a synthetic growth defect with unknown off-target binding cannot be excluded.

*I still think some data are required to show that compound 3 dissociates BD2-deficient Bdf1 from chromatin in cells. Such an experiment may be done in *C. albicans*, *S. cerevisiae*, or related organisms. It could be monitored by ChIP, biochemical fractionation, fluorescent imaging with a proper tag, or other methods. Transient expression systems may also be used.*

Indeed, such additional data would certainly have enhanced our results and we regret not having been able to provide these data in our revised manuscript. With respect to the comment that BD2-deficient strains are more unhealthy than BD1-deficient strains, we note that Figure 8e (formerly Figure 4e) shows that the two strains bearing a single point mutant in BD2 (*bdf1-Y425F/pMET-BDF1* and *bdf1-Y425F/pTetO-BDF1*) grow essentially as well as the corresponding strains in which BD1 has been deleted (*bdf1-bd1Δ/pMET-BDF1* and *bdf1-bd1Δ/pTetO-BDF1*). Whereas the strains mutated in BD2 show a dramatic decrease in growth upon addition of compound **3**, no significant effect is seen for the strains deleted for BD1, strongly suggesting that the inhibitory effect of **3** is mediated by its interaction with the functional BD1 domain in the BD2-deficient strains.

The other issues were properly addressed in the revised manuscript. As a minor point, several lines seem to be missing in Supplementary Table 1.

Thanks to the reviewer for pointing this out. The missing lines have been added in the newly revised manuscript.

Reviewer #3:

The authors have addressed all minor issues previously pointed out.

We are pleased that the Reviewer is satisfied with the revisions.